**EMBO** *reports*

# Vesicle-coupled mRNA transport and translation govern intracellular organelle networking

Melissa Vázquez-Carrada [iD] [1,4], Sainath Shanmugasundaram [iD] [1,4], Sander H J Smits [iD] [2,3], Lasse van Wijlick[1] & Michael Feldbrügge [iD] [1✉]

## Abstract

**Eukaryotic cells are highly compartmentalized, enabling sophisticated division of labour. For example, genetic information is stored in the nucleus while energy is produced in mitochondria. Despite this clear specialisation, compartments depend on intensive communication, including the exchange of metabolites and macromolecules. This is achieved through intracellular trafficking with membranous carriers such as endosomes, which constitute versatile transport vehicles. Key cargos include mRNAs and ribosomes that hitchhike on endosomes, linking RNA and membrane biology. In this review, we summarize recent advances showing how mRNAs are mechanistically attached to membranes of endosomes and lysosomal vesicles and how cargos are identified for transport. The encoded proteins illuminate the biological processes that rely on such spatiotemporal control. This is particularly true for the regulation of subcellular mitochondrial homeostasis, disclosing intensive multi-organelle networking. As a general concept, the underlying protein/protein and protein/RNA interactions exhibit significant redundancy yet are organized in a strict hierarchy with distinct core and accessory functions. This ensures both the robustness and specificity of mRNA hitchhiking.**

**Keywords** Endosomes; Local Translation; Mitochondria; RNA Transport; SLiMs
**Subject Categories** Cell Adhesion, Polarity & Cytoskeleton; Membranes & Trafficking; Translation & Protein Quality

## Introduction

The evolutionary success of eukaryotic cells is based on extensive compartmentalization, allowing a complex division of labor by distinct organelles. A characteristic of eukaryotes is the nucleus, which serves as a storage place for genetic information in DNA and as a regulatory hub for the transcription of pre-mRNAs. Cytoplasmic entities include the endoplasmic reticulum (ER), which organizes the secretory pathway, and mitochondria, which

function as the powerhouse of the cell. These compartments are not isolated units executing their function independently, but they form an intricate intracellular network for the exchange of macromolecules and metabolites. One of the best examples of macromolecule trafficking is the fact that more than 99% of the mitochondrial proteins are encoded by nuclear genes (Bykov et al, 2020). Therefore, the resulting translation products must be delivered to mitochondria in a highly organized and controlled fashion. Their mistargeting in the cytoplasm has been linked to human diseases and aging (Kaushik et al, 2025).

In recent years, it has become evident that transport of mRNAs and their local translation are key processes to support intracellular networking linking RNA and membrane trafficking (Béthune et al, 2019). Local translation at the rough ER (rough, because of the presence of ribosomes) was reported early on (Blobel and Dobberstein, 1975); however, more recently evidence is accumulating that local translation occurs at numerous intracellular membranes like those of mitochondria, peroxisomes, early endosomes, late endosomes and secretory vesicles (Baumann et al, 2014; Christensen and Reck-Peterson, 2022; Cioni et al, 2019). Thereby, import into organelles such as rER or mitochondria, as well as co-translational membrane attachment of translation products on the cytosolic surface of organelles, is promoted (see below). A common mechanism to enable translation in the vicinity of membranes is the function of RNA-binding proteins (RBPs) for mRNA delivery and anchoring, facilitated by membrane-associated adapter proteins.

An emerging research field of intracellular trafficking is vesicle-coupled transport and translation of mRNAs that mediate the organization of polarity factors and cytoskeletal elements, besides the delivery of proteins to mitochondria to support their function (Müntjes et al, 2021; Zilio et al, 2025; Fig. 1A,B). In this review, we describe novel advances addressing outstanding questions: how are mRNAs linked to endosomes? How are mRNAs selected for transportation and/or local translation? Addressing these questions provides information about the cell biological processes that are particularly dependent on this translocation mechanism. A special focus is placed on the role of endosomal transport during polar growth of infectious hyphae from the corn pathogen *Ustilago maydis*, a model system that has provided foundational insights for the field (Fig. 1A). Comparison of *U. maydis* with other systems, from plants to vertebrate neurons (Fig. 1B), reveals fundamental principles of organelle networking that are found throughout the

[1]Department of Biology, Institute of Microbiology, Cluster of Excellence on Plant Sciences, Heinrich Heine University Düsseldorf, Düsseldorf 40204, Germany. [2]Center for Structural Studies, Heinrich Heine University Düsseldorf, Düsseldorf 40204, Germany. [3]Department of Chemistry, Institute of Biochemistry, Heinrich Heine University Düsseldorf, Düsseldorf 40204, Germany. [4]These authors contributed equally: Melissa Vázquez-Carrada, Sainath Shanmugasundaram. ✉E-mail: feldbrue@hhu.de

**Glossary**

| | |
|---|---|
| ALG2 | Apoptosis linked gene 2 |
| ALS | Amyotrophic lateral sclerosis |
| AKAP1 | Protein A-kinase anchoring protein 1 |
| ATP | Adenosine triphosphate |
| BORC | BLOC1-related complex |
| CNBP | CCHC-type zinc finger nucleic acid binding protein (CNBP) |
| Cdc | Cell division control protein |
| CORVET | Class C core vacuole/endosome tethering |
| EEA1 | Early endosome antigen 1 |
| ELAV | Embryonic lethal abnormal vision |
| ER | Endoplasmic reticulum |
| ETC | Electron transport chain |
| FERRY | Five-subunit endosomal Rab5 and RNA/ribosome intermediary |
| FYVE | Fab 1, YOTB, Vac 1, and EEA1 |
| GEF | Guanine nucleotide exchange factor |
| GTP | Guanosine triphosphate |
| iCLIP2 | Individual-nucleotide resolution UV crosslinking and immunoprecipitation 2 |
| IDRs | Intrinsically disordered regions |
| LAMP1 | Lysosomal-associated membrane protein 1 |
| LARP4 | La-related protein 4 |
| LOCL-TL | LOV-domain-controlled ligase for translation localization |
| m6A | N6-methyladenosine |
| MIRO | Mitochondrial Rho |
| MLLE | MademoiseLLE domain |
| mRNA | Messenger RNA |
| mRNPs | Messenger ribonucleoproteins |
| NAC | Nascent chain-associated complex |
| NSF | N-ethylmaleimide-sensitive factor |
| PAM2 | PAB1C-associated motif |
| PI3P | Phosphatidylinositol 3-phosphate |
| PI(3,5)P$_2$ | Phosphatidylinositol 3,5-bisphosphate |
| PINK1 | PTEN-induced serine/threonine kinase 1 |
| RBD | RNA-binding domain |
| RBP | RNA-binding protein |
| ROCs | Rotamase cyclophilins |
| rER | rough endoplasmic reticulum |
| RRMs | RNA recognition motifs |
| SLiM | Short linear motif |
| STRIPAK | Striatin-interacting phosphatase and kinase complex |
| TRPML | Transient receptor potential cation channel, mucolipin subfamily |
| TOM | Translocase of the mitochondrial outer membrane |
| UTR | Untranslated region |
| VIT | Very important target |

three kingdoms of eukaryotes. The process is widespread and highlights the intimate organelle networking that relies on the link between RNA and membrane biology (Table 1). This holds particularly true for the flow of genetic information along a critical communication axis: from the nucleus, via endosomes, to mitochondria. This pathway represents a central and common solution for maintaining mitochondrial proteostasis and function at cellular outposts, underscoring the intimate and indispensable link between RNA and membrane biology. For deeper insights into specific aspects of mRNA trafficking or mitochondrial protein import, we refer to excellent recent reviews (Bourke et al, 2023; Chekulaeva, 2024; Sharma and Fazal, 2024; Vargas et al, 2022; Zilio et al, 2025, see also Box 1 for mitochondrial mRNA hitchhiking).

## Basic concepts of organelle-coupled transport and translation of mRNAs

Precise local translation of mRNAs can ensure the correct subcellular destination of the encoded proteins. The underlying accumulation of mRNAs at specific subcellular sites can be achieved by different mechanisms, such as regulation of mRNA stability or by active, motor-dependent mRNA transport (Cheku-laeva, 2024). Stability-driven localization has been shown during development in *Drosophila melanogaster*. *nanos* mRNA, for example, is specifically protected at the posterior pole of embryos. This results in polar translation triggering further posterior development (Zaessinger et al, 2006). In neurons, mRNAs associated with housekeeping functions localize specifically to neurites because at this location their stability is increased due to protection from microRNA-mediated degradation or altered m6A modification (Loedige et al, 2023; Mendonsa et al, 2023). This stability-driven mechanism supports a basal level of protein production throughout the neuron, ensuring that essential functions can be maintained far from the cell body.

Alternatively, active transport along the cytoskeleton serves as a prevalent mode of trafficking for mRNA localization and local translation. Translocation of cargo mRNAs is mediated by molecular motors such as myosin for actin-dependent short-distance transport or the interplay of kinesin and cytoplasmic dynein for long-distance shuttling along microtubules (Bourke et al, 2023; Chekulaeva, 2024; Mofatteh and Bullock, 2017; Moissoglu et al, 2025). Noteworthy, the two strategies for mRNA localization, i.e., either stability-driven or motor-mediated, are not mutually exclusive. Certain transcripts might be actively transported to subcellular areas of higher or lower mRNA turnover.

At present, different mechanisms of translocation for active mRNA transport are known that differ in their complexity. The most straightforward strategy is linking mRNA cargo-binding RBPs either directly or indirectly via adapter proteins to molecular motors. A well-known example is the She2/She3 RBP complex connected to myosin for actin-dependent trafficking. Thereby, cargo mRNA is moved to the distal pole of daughter cells during cell division in *S. cerevisiae* (Niessing et al, 2018). A more recent example reports that the RBP CNBP (CCHC-type zinc finger nucleic acid binding protein) directly interacts with KIF1C kinesin for microtubule-dependent transport in protrusions of mammalian cells (Moissoglu et al, 2025). In neurons, the RNA-binding APC (adenomatous polyposis coli) interacts with the KAP3 cargo loading subunit of kinesin KIF3A/B for axonal localization of target mRNAs (Jimbo et al, 2002; Preitner et al, 2014; Ruane et al, 2016). To study the basics of this canonical mRNA transport mechanism, sophisticated in vitro systems are used that reproduce the activity of the core machinery in action. This core consists of transport RBPs, mRNAs containing cognate RBP binding sites, as well as adapters and molecular motors (Baumann et al, 2020; Edelmann et al, 2017; Heber et al, 2024; McClintock et al, 2018).

However, in a cellular environment, the mode of transport for numerous mRNAs is most likely mediated by higher-order mRNP complexes consisting of multiple RBPs coordinating translation

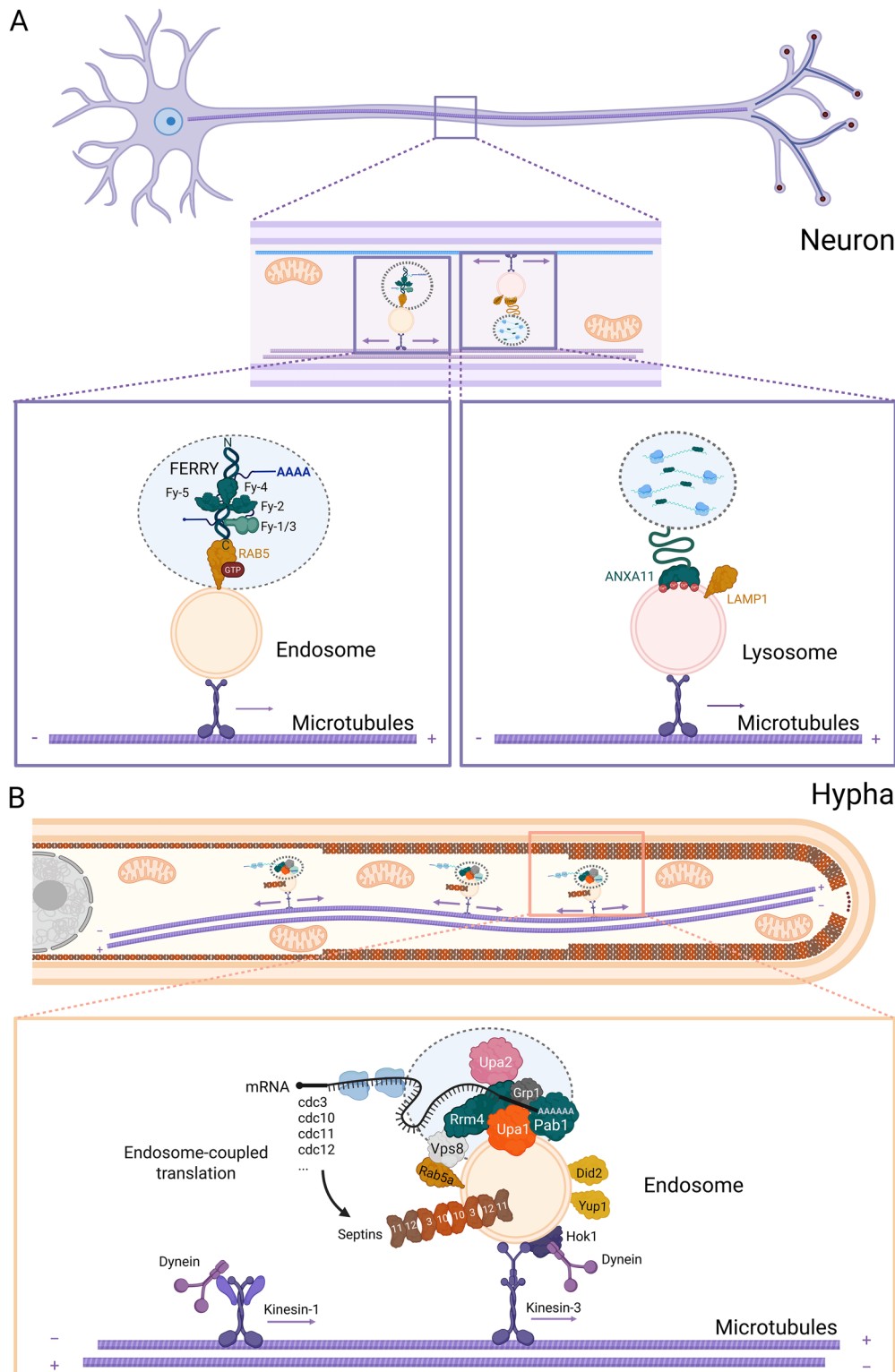

**Figure 1. Vesicle-coupled mRNA transport in polarized cells.**

Overview of vesicle hitchhiking in (**A**) a mammalian neuron (top) and (**B**) a fungal hypha (bottom). The boxes offer a magnified view of the different transport mechanisms. The five-subunit FERRY complex links to endosomes via an interaction with RAB5 (upper left). In a distinct mechanism, the ANXA11 adapter links larger RNA granules to LAMP1-positive lysosomes using its N-terminal intrinsically disordered region (upper right). In fungi, the core RNA-binding protein Rrm4 is linked via anchor proteins to the endosomal surface. As depicted in the hypha, endosome-coupled translation of all four septin mRNAs is essential for the assembly of heteromeric septin complexes, which are subsequently transported to form higher-order filaments at the growth pole (left; further details are given in the text; created in https://BioRender.com).

**Table 1. Comparison of vesicle-dependent mRNA translocation machineries**

| | Rrm4 machinery | FERRY machinery | ANXA11 machinery | RBP-L/RBP-P machinery | ROC machinery |
|---|---|---|---|---|---|
| Key RBP | Rrm4 | Fey2 with noncanonical RNA binding | ANXA11 N-terminus | RBP-L and RBP-P RRM domains | Noncanonical rotamase cyclophilin, ROC3 and ROC5 |
| Organism | *Ustilago maydis* | *Homo sapiens* (HeLa and neuronal cells) | *Homo sapiens / Danio rerio* | *Oryza sativa* | *Arabidopsis thaliana* |
| Vesicle type | Early endosome, Rab5a positive | Early endosome, RAB5 positive | Lysosome, LAMP1-positive | Early endosome, Rab5 positive | Early endosome, Rab5a positive |
| Vesicle adapter | Upa1 and Vps8/Rab5a | Fey2 and Rab5 | ANXA11 C-terminal repeated region | RBL-L and Rab5 | Rab5 |
| Key mRNAs | Hundreds of mRNAs included the ones that encode septins and mitochondrial ETC proteins | About 250 mRNAs included the ones that encode mitochondrial proteins | β-actin and ribosomal proteins, mitochondrial ETC proteins | glutelin, prolamin storage protein | Flowering locus |
| Technical approach | iCLIP and iCLIP2 | GST pull-down | Microfluidics and RNA seq of transport mutants | Co-purification | RIP analysis |
| Global biological function | Hyphae polar growth/plant infection | Long-distance transport (Proposed) | Neuronal functions/ALS neuronal disease | Endosperm development | Flowering |
| Specific function | Septin assembly | mRNA transport (Proposed) | RNA granules transport | Glutelin mRNA transport to the cortical ER | Mobile mRNAs intracellular transport to plasmodesmata |
| | Bulk mRNA and ribosome distribution | | β-actin mRNA transport, cytoskeleton | | |
| | Specific mRNA regulons transport | | mRNAs encoding ETC complexes transport | | |
| | Mitochondrial protein import | | Mitochondrial and axonal homeostasis | | |
| | Mitochondrial homeostasis | | | | |
| | Endosome- coupled translation loads the surface with newly synthesized proteins (e.g., septins) | | | | |
| | Cell wall remodeling | | | | |
| References | Müntjes et al, 2021; Stoffel et al 2025b; Devan et al 2024; Olgeiser et al, 2019; König et al, 2009 | Schuhmacher et al, 2023; Quentin et al 2023 | Liao et al, 2019; Nixon-Abell et al 2025; dePace et al 2024 | Tian et al, 2020a | Luo et al, 2024 |

**Box 1   mRNA hitchhiking on the mitochondrial surface**

Beyond their metabolic roles, mitochondria can act as transport vehicles for specific mRNAs, a mechanism known as mitochondrial hitchhiking. This process offers an attractive alternative to vesicular transport, ensuring that proteins destined for mitochondria are synthesized precisely where they are needed within the cell (Cohen et al, 2024). The *COX7c* mRNA, for example, encoding a critical subunit of the mitochondrial electron transport chain, is co-transported with mitochondria along axons through a coding-region-dependent mechanism. Thereby, essential metabolic proteins can be synthesized on demand in energy-hungry regions like neuronal synapses (Cohen et al, 2022). A more complex example is the journey of the mRNA encoding PINK1 (PTEN-induced kinase 1), a critical initiator of mitophagy. This process unfolds as a sophisticated cycle. It begins when the RNA-binding protein SYNJ2 binds to the *PINK1* mRNA and docks the entire complex onto the mitochondrial outer membrane via the anchor protein SYN2BP. Once attached, the mRNA "hitchhikes" on the mitochondrion as it travels to distant sites, for example, in neurites (Harbauer et al, 2022). This association is tightly controlled; in response to insulin signaling, SYN2BP is dephosphorylated, triggering the release of the *PINK1* mRNA. Thereby, the mRNA is available for local translation to activate mitophagy (Hees et al, 2024b). Following translation, the mRNA can associate with other organelles, such as the ER, from which it can be re-routed back to the mitochondria through the ER-SURF pathway, completing a highly regulated trafficking loop (preprint: Hees et al, 2024a). Mitochondrial hitchhiking also plays a crucial role in cellular architecture and spatial awareness. Mitochondrial transport along microtubules is driven by the TRAK2-MIRO1 complex, with TRAK2 acting as an adapter linking the organelle to motor proteins for movement (Canty et al, 2023; Fenton et al, 2021). Remarkably, the transport of *TRAK2* mRNA itself on the mitochondrial surface is proposed to function as a "molecular ruler." This process provides the cell with spatial information about its own geometry, helping to maintain organellar homeostasis across different cell sizes and shapes (Bradbury et al, 2025). In essence, mitochondrial mRNA hitchhiking might not just act as a simple delivery system but function as an integral part of the cell's machinery for spatial sensing and self-organization.

**Box 2   In need of answers**

(i) How are mRNPs precisely linked to endosomes? Structure-function analyses of mRNP-endosome complexes are needed.

(ii) How are mRNAs selected to form transport mRNPs, and how are these loaded and unloaded from transport vesicles? Insights into key RBPs, their interaction partners, and how these factors are post-translationally regulated are required.

(iii) How is the link between vesicles and mRNPs dynamically modulated? We need to study key posttranslational modifications in RBPs, changes in calcium homeostasis, and/or lipid identity.

(iv) How does membrane-coupled translation operate at the subcellular level under physiological conditions? Biosensors and optogenetics can be used to study the influence of localized translation.

(v) How is membrane-coupled localized translation linked to biomolecular condensates?

(vi) How widespread is membrane-coupled translation? Fundamental principles from different systems like cyanobacteria, fungi, plants, and humans need to be compared.

(vii) How are different active transport mechanisms, such as classical RBP/motor-based systems or vesicle-coupled systems, coordinated in the same cellular environment, such as in neurons?

Co-transport of vesicles and mRNPs was initially discovered to be essential for efficient polar growth of infectious hyphae in *U. maydis* (Baumann et al, 2012; Higuchi et al, 2014; König et al, 2009). Since then, vesicle hitchhiking has also been found in other fungi as well as in plants, animals, and humans (Cioni et al, 2019; De Pace et al, 2024; Liao et al, 2019; Luo et al, 2024; Stein et al, 2020). Biological processes that depend on this mRNA translocation mechanism cover a wide range, such as fungal infection, endosperm development in plants, and neuronal plasticity and memory in humans (see below and Table 1). Unresolved questions in the field are how cargo mRNAs are attached to endosomes and how they are selected for transport (see Box 2, for additional questions in need of answers).

## The role of short linear motifs in the association of mRNPs with the cytosolic surface of vesicles

According to the basics of endosomal mRNA trafficking, key RBPs recognize specific RNA elements within cargo mRNAs and interact with adapter proteins on the cytoplasmic surface of endosomes. These endosomal proteins determine the specificity for the vesicle system of choice. In plants, for example, the RRM protein RBP-L interacts with the GTP-bound form of the small GTPase Rab5 that serves as a membrane marker for early endosomes (Tian et al, 2020b; see below).

However, a simple RBP/adapter complex is not sufficient for full functionality, and a more complex protein interaction network is needed (Fig. 2). A common concept for the formation of such sophisticated RBP networks is the recognition of short linear motifs, designated SLiMs, which often occur within intrinsically disordered regions of interaction partners and serve as contact points for multivalent low-affinity interactions (He et al, 2023). The C-terminal MademoiseLLE (MLLE) domain of the human cytoplasmic poly(A)-binding protein PABPC1 constitutes a SLiM recognition domain consisting of a defined five-helix architecture

and transport. These can even form large ribonucleoprotein granules in membrane-less biomolecular condensates through phase separation (Brangwynne et al, 2009; Kato and McKnight, 2018). Key factors are multivalent interactions mediated by RNA and protein partners containing intrinsically disordered regions (IDRs) that govern their assembly for transport (Fernandopulle et al, 2021; Protter and Parker, 2016). Examples of such sophisticated transport granules include (i) localization bodies in oocytes from *Xenopus laevis* transporting numerous RNAs to the vegetal pole (Neil et al, 2021) or (ii) *oskar* ribonucleoprotein granules in oocytes from *D. melanogaster*, which depend on architectural RNA-RNA interactions and exhibit a critical liquid- to solid-like phase transition (Bose et al, 2022; Bose et al, 2024).

Another prominent mRNA translocation mechanism using defined motors and adapters is vesicle-associated hitchhiking (Table 1). This translocation mechanism was found to transport a variety of cargos, such as lipid droplets and peroxisomes (Christensen and Reck-Peterson, 2022). Importantly, mRNAs and ribosomes also constitute prominent cargos and link this process to mRNA trafficking, explaining how hundreds of mRNAs can be transported with a limited set of motor and adapter proteins (Pushpalatha and Besse, 2019). Even membrane-less RNP granules can hitchhike on membrane-enclosed vesicles for transport (Liao et al, 2019; Pushpalatha and Besse, 2019).

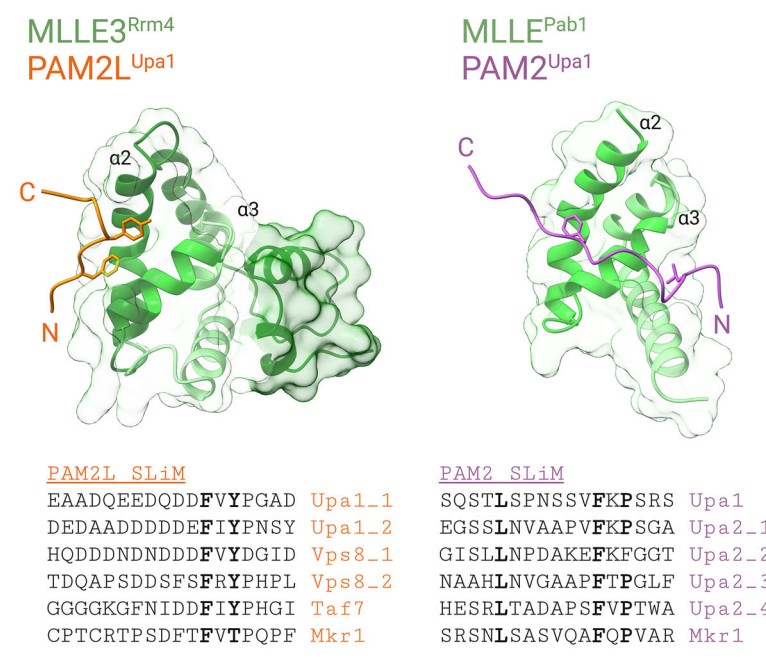

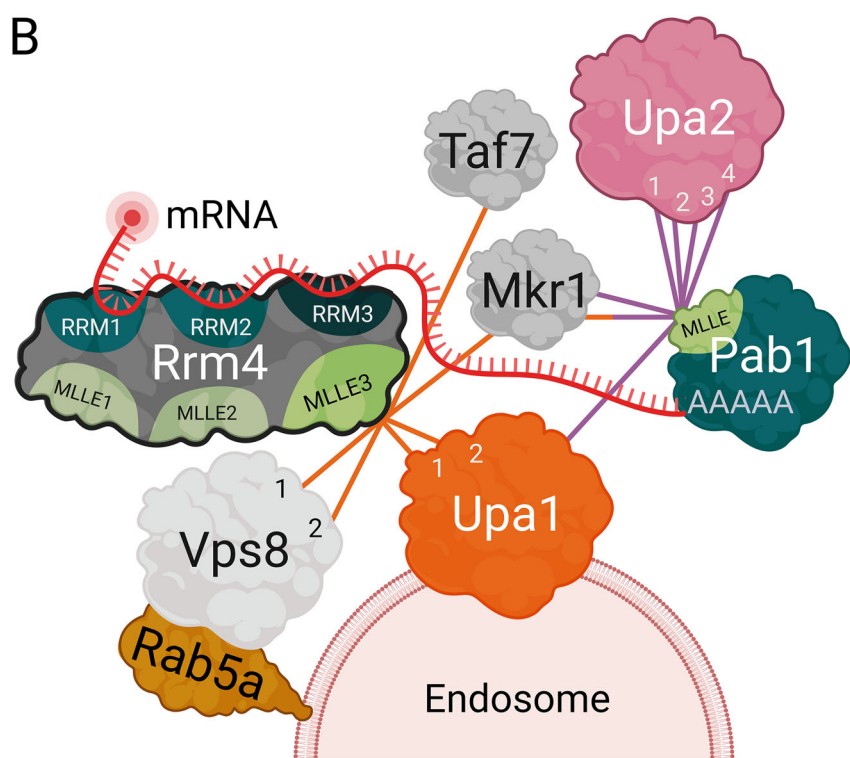

**Figure 2. A sophisticated Mademoiselle/SLiM network orchestrates endosomal attachment of mRNPs during transport.**

(A) Experimentally determined co-structures of two distinct MLLE domains from *U. maydis*. The novel seven-helix MLLE3 domain of the RNA-binding protein Rrm4 (additional two helices in dark green) recognizes a PAM2-like SLiM (orange), whereas the canonical five-helix MLLE domain of the poly(A)-binding protein Pab1 binds a PAM2 SLiM (purple). The critical helices α2 and α3 that form the binding interface are indicated. Below, the sequences of PAM2L and PAM2 SLiMs mediating the interaction with their respective MLLE domains are listed. (B) A model depicting the intricate interaction network that assembles on the endosomal surface to facilitate mRNP transport. The MLLE domains of core transport proteins like Rrm4 and Pab1 interact with a variety of SLiM-containing adapter proteins (e.g., Upa1 and Upa2), linking cargo mRNAs to the endosomal membrane and coordinating the transport machinery (interactions depending on PAM2 or PAM2-like SLiMs are indicated by purple or orange lines, respectively; see text for further details; created in https://BioRender.com).

(Kozlov et al, 2002; Kozlov et al, 2001). Helices 2, 3, and 5 form a distinct binding interface for specific recognition of a defined SLiM, designated PAM2 (PABPC1-associated motif; Fig. 2A), in PABPC1 binding partners. Through this mechanism, PABPC1 interacts with numerous partners such as GW182, eRF3, and LARP4, functioning in microRNA regulation, translational termination, and posttranscriptional control, respectively (Jinek et al, 2010; Kozlov and Gehring, 2010; Yang et al, 2011). However, these experimentally verified interactions are most likely only the tip of the iceberg, since 77 of 120 human PABPC1 interactors, which were determined by in vivo proximity labeling (Youn et al, 2018), contain PAM2 or slightly related SLiMs (He et al, 2023).

The importance of SLiM networking during endosomal mRNA transport became apparent by studying the key mRNA transporter Rrm4 from *U. maydis*. This central RBP contains three N-terminal RRM domains for cargo mRNA recognition (see below) and three C-terminal MLLE domains forming a defined interaction platform with a strict hierarchy (Devan et al, 2022; Devan et al, 2024): MLLE3 is necessary and sufficient for endosomal attachment, while MLLE1 and MLLE2 - with currently unknown cognate binding motifs - carry out accessory functions (Fig. 2B).

Although MLLE3 of Rrm4 is highly similar to its counterpart in Pab1, it does not bind PAM2 sequences. Instead, it recognizes a related SLiM sequence, called PAM2-like (PAM2L), with exquisite sequence specificity. Experimental analysis of the peptide-bound MLLE structure deciphered the underlying binding code (Devan et al, 2024): MLLE3 forms a novel type of seven-helix domain with a noncanonical interaction interface along helix 2, contacting a critical conserved tyrosine that is not present in PAM2 motifs. The two additional helices do not provide direct contact but participate in shaping the noncanonical binding pocket (Fig. 2A).

Two functionally redundant PAM2L sequences for interaction with MLLE3 of Rrm4 are present in the endosomal adapter protein Upa1 (Fig. 2A,B; Pohlmann et al, 2015). Upa1 contains a FYVE zinc finger domain for specific recognition of phosphatidylinositol 3-phosphate (PI3P) lipids, which are characteristic of early endosomes (Pohlmann et al, 2015). Loss of Upa1 causes aberrant motility of Rrm4 and severe defects in unipolar hyphal growth (Pohlmann et al, 2015). However, in the absence of Upa1, Rrm4 still associates with endosomes but less efficiently. Hence, although Upa1 is a key adapter, redundant interaction partners for endosomal attachment must be present. Decoding the PAM2L code allowed de novo prediction and experimental verification of additional MLLE3 interaction partners. For example, Vps8 contains two PAM2L motifs at its N-terminus and functions during endosome maturation by recruiting the evolutionarily conserved CORVET complex (class C core vacuole/endosome tethering; Devan et al, 2024; Perini et al, 2014; Schneider et al, 2022). Expression of an N-terminally truncated version of Vps8, missing both PAM2L sequences, resulted in aberrant shuttling of Rrm4-positive endosomes accumulating at the growth pole (Devan et al, 2024). Thus, although Vps8 is not the missing endosomal adapter, we concluded that the SLiM-based interaction network carries out various functions related to endosomal activity (Devan et al, 2024; Fig. 2B). The presence of extensive SLiM-based interactions is further supported by the observation that another core component, Upa2, serving as a scaffold protein for endosomal mRNP assembly, contains four redundant PAM2 sequences for interaction with Pab1 (Fig. 2B; Jankowski et al, 2019).

MLLE-based SLiM interaction networks are not restricted to *U. maydis*. In humans, the two MLLE-containing proteins PABPC1 and HECT chain-elongating E3 ubiquitin ligase UBR5 (Homologous to E6AP C-Terminus; Hehl et al, 2024) differentiate between various binding partners, as for example the RNA-binding E3 ubiquitin ligase MKRN1 (Makorin, Devan et al, 2024) as previously hypothesized (He et al, 2023). Thus, comparable SLiM-based networks are most likely also operational in other vesicle-mediated mRNA transport systems.

# Alternative mechanisms for the association of mRNPs with the cytosolic surface of vesicles

A common theme in endosomal attachment of mRNPs is the interaction with the small G-protein Rab5a. Vps8, for example, is known to directly interact with Rab5a as part of the CORVET complex (Schneider et al, 2022). In plants, it has been described that the two RRM proteins, RBP-P and RBP-L, interact with the endosomal component NSF (*N*-ethylmaleimide-sensitive factor; Tian et al, 2020a) and GTPase Rab5, forming a quaternary complex containing cargo mRNAs (Tian et al, 2020a). Moreover, rotamase cyclophilins have been identified as noncanonical RBPs involved in endosomal hitchhiking for selective transport of mRNAs to plasmodesmata for cell-to-cell passage. However, neither their mode of RNA binding nor the membrane attachment of these interesting new factors is currently known (Luo et al, 2024; Table 1).

In humans, structural analysis revealed that the FERRY complex (Five-subunit Endosomal Rab5 and RNA/ribosome intermediarY) functions as RAB5 effector (Quentin et al, 2023; Schuhmacher et al, 2023). Remarkably, mutations in the FERRY complex have been implicated in neurological disorders (Riffe & Downes, 2025). The scaffold subunit FY2 links FERRY members and mRNAs to RAB5. However, RNA binding does not involve a canonical RNA-binding domain but rather a clamp-like structure. Sequence-specific binding has not yet been elucidated, but bound mRNAs encode regulons involved in polarity and mitochondrial homeostasis (Schuhmacher et al, 2023). Based on the observation that the FERRY complex co-localizes with early endosomes in neurons, the authors hypothesize a function during endosomal mRNA transport and mitochondrial protein import (Quentin et al, 2023; Schuhmacher et al, 2023; see below, Table 1).

As mentioned above, membrane-less biomolecular condensates can also be coupled to membranes. The annexin ANXA11 links mRNA-containing condensates, so-called RNA granules, to lysosomal LAMP1-positive vesicles during transport in axons (Liao et al, 2019; Fig. 1). The adapter contains an N-terminal disorder region for RNA granule formation and a C-terminal repeat region for membrane attachment. The latter recognizes specific PI(3,5)P$_2$ lipids in lysosomal membranes in a Ca$^{2+}$-dependent manner. Hence, local Ca$^{2+}$ release mediated by the endolysosome-localized mucolipin transient receptor TRPML (Dong et al, 2010) might determine RNA granule association (Liao et al, 2019). High activity might promote loading, and low activity could trigger unloading of RNA granules, an attractive mechanism explaining delivery. Consistently, a TRPML agonist has been reported to increase ANXA11 recruitment (Liao et al, 2019). Alternatively, local activity of PI(3,5)P2 dynamics mediated by e.g. PI5P PIKfyve kinase or the

corresponding 5-phosphatase might modulate RNA granule association (Liao et al, 2019). Interestingly, the association of ANXA11-containing condensates triggers a liquid-to-gel phase transition in the juxtaposed lipids, suggesting a functional cross-talk between biomolecular condensates (Nixon-Abell et al, 2025). This process can be modulated by ANXA11 interaction partners such as mannosyltransferase ALG2 and calcyclin CALC, and thereby also influences ANXA11-dependent condensation. ALG2, for example, promotes condensation and stiffening of the system. This more rigid conformation might be important during active transport. In contrast, CALC functions as an antagonist, decreasing ANXA11 condensation. This elicits softening, which might favor loading and unloading of RNA granules (Nixon-Abell et al, 2025). Interestingly, ANXA11-mediated trafficking maintains mitochondrial homeostasis in axons and counteracts their degeneration (De Pace et al, 2024). Thus, mRNA transport mediated by endosomal and lysosomal vesicles appears to be linked to mitochondrial functions (see below; Table 1; De Pace et al, 2024; Quentin et al, 2023; Schuhmacher et al, 2023).

## Recruitment of mRNAs for vesicle-coupled transport

A critical question is how mRNAs are selected for transport. Individual mRNAs can be translocated by RBPs using canonical RNA-binding domains to specifically recognize RNA transport elements in cargo mRNAs (Das et al, 2021). The She2/She3 RBP complex, for example, recognizes four transport elements in *ASH1* mRNA for actin-dependent trafficking (Niessing et al, 2018). Furthermore, the RNA-binding protein CNBP interacts with GA-rich sequences in cargo mRNAs for microtubule-dependent transport towards cellular protrusions (Moissoglu et al, 2025). However, in highly polarized cells such as neurites and hyphae, numerous mRNAs are transported simultaneously and organized in functional RNA regulons (De Pace et al, 2024; Liao et al, 2019; Olgeiser et al, 2019; preprint: Stoffel et al, 2025b). Thus, a central challenge is the identification of key transport RBPs and the transcriptome-wide elucidation of how they use their RNA-binding domains (RBDs) to recognize functionally important binding sites in cargo mRNAs. Notably, RBPs often contain multiple RBDs to coordinate transport. The human key transport RBP Staufen2, for example, contains four double-stranded RBDs that contribute differentially to RNA binding (Heber et al, 2019).

A recent breakthrough was achieved in studying the ELAV-type RNA transporter Rrm4 containing three N-terminal RRM domains (preprint: Stoffel et al, 2025b). The tandem RRM1 and RRM2 are essential for efficient unipolar growth in *U. maydis*, whereas RRM3 is dispensable (Becht et al, 2006). The underlying RRM binding code was deciphered by comparing RNA recognition of mutants in each RRM domain to the wild-type (preprint: Stoffel et al, 2025b). To also detect weak RNA binding of mutants, the improved iCLIP2 procedure (Buchbender et al, 2020) was adapted to microorganisms with high intrinsic protease and RNAse activity, resulting in a tailor-made microbial iCLIP2 procedure (Stoffel et al, 2025a).

The mutant-based comparative approach revealed more than 50,000 binding sites in thousands of mRNAs. More than 11,000 binding sites were RRM3-responsive, and about 80% contained the sequence UAUG. However, since RRM3 is dispensable for unipolar

hyphal growth, the simple binary interaction RRM3/UAUG is most likely not needed to foster polar growth. This interaction functions most likely during bulk mRNA transport (preprint: Stoffel et al, 2025b). Instead, functionally important composite binding sites exhibit a more complex recognition mode involving all three RRMs: RRM1 and RRM2 provide specificity, whereas RRM3 supports binding (preprint: Stoffel et al, 2025b). So far, only RRM3 binding to UAUG could be reconstituted in vitro, suggesting the involvement of additional components during the recognition of complex functional RNA localization elements. Interestingly, loss of Rrm4 resulted in a reduced amount of those mRNAs containing functional binding sites, suggesting reduced mRNA stability. Thus, Rrm4 could play dual roles in cytosolic stability and endosomal transport. Alternatively, Rrm4 target mRNAs might be protected from degradation during transport, linking mRNA stability with translocation (preprint: Stoffel et al, 2025b). This hypothesis was also proposed in neurons due to the duration of endosome movement over long distances. Consistently, loss of FERRY subunits resulted in decreased levels of target mRNAs (Schuhmacher et al, 2023).

Work on Rrm4 has revealed two different transport strategies. The simple binary interaction RRM3/UAUG is most likely needed for global mRNA distribution throughout the hyphae, whereas a more complex binding mode involving RRM1 and RRM2 with the accessory function of RRM3 is needed to recognize functional elements. The comprehensive mutant-based in vivo RNA-binding analysis of the mRNA transporter Rrm4 might serve as a blueprint for the analysis of other RBPs.

In other systems, the recruitment of mRNAs is currently intensively studied. Key components for endosomal association with RNA-binding capacity have been identified, such as the human ANXA11 or FERRY complex (Liao et al, 2019; Quentin et al, 2023; Schuhmacher et al, 2023). For the latter, about 250 different transcripts were identified by co-purification containing potential cargo mRNAs with functions in the regulation of polarity and mitochondrial homeostasis (Schuhmacher et al, 2023; see below). ANXA11 associates with RNA stress granules, sharing important RBPs such as FUS and hnRNPA3 (Liao et al, 2019). However, specificity for cargo selection, such as β-actin mRNA, is less clear. In plants, the two RRM proteins, RBP-P and RBP-L, are crucial for endosomal hitchhiking; however, their precise mode of RNA binding, as well as a transcriptome-wide view, is currently missing (Tian et al, 2020a).

Taken together, uncovering how Rrm4 uses its three RRM domains to recognize different types of RNA transport elements is one better understood example of how the translocation of hundreds of mRNAs can be coordinated. However, it is conceivable that in other systems, such as neurons, RNA granule-mediated global trafficking of cargo mRNAs lacking specific RNA transport elements is coupled to localization-driven stability (Chekulaeva, 2024) to achieve precise subcellular localization of mRNAs.

## Biological functions of vesicle-mediated mRNA transport in fungi

In fungi, vesicle-mediated mRNA transport is needed for polar growth of hyphae. One prominent example is the role of endosomal mRNA transport during the formation of unipolar infectious

hyphae of *U. maydis* (Müntjes et al, 2021). This links endosomal transport to infection of its host plant, corn, during smut disease. Loss of endosomal transport results in the formation of aberrant bipolar hyphae and reduced growth (Becht et al, 2006).

Essential steps of the hyphal growth program are the local deposition and activation of polarity factors, such as small G proteins that elicit the reorganization of the actin and septin cytoskeleton. Thereby, cell wall remodeling enzymes and building blocks are delivered to the growth pole for efficient apical expansion of membrane and cell wall (Riquelme et al, 2018). Analysing the cargo mRNAs of endosomal transport during polar growth revealed two distinct transport strategies (i) transport of distinct sets of mRNAs with functions in hyphal growth and (ii) global transport of bulk mRNAs and associated ribosomes (Higuchi et al, 2014; König et al, 2009; Olgeiser et al, 2019; preprint: Stoffel et al, 2025b).

As pointed out above, this requires a sophisticated recognition code of the central RNA transporter Rrm4 that uses a hierarchical system where multiple RNA-binding domains cooperate to differentiate between high-priority functional cargo and bulk transport (see above, Olgeiser et al, 2019; Stoffel et al, 2025a; preprint: Stoffel et al, 2025b).

Even distribution of bulk mRNAs and associated ribosomes avoids gradients of mRNAs, translation products, and ribosomes in the cell. Hence, endosomal hitchhiking of translationally active mRNAs in the form of polysomes provides an elegant mechanism to transport ribosomes over long distances in *U. maydis* (Baumann et al, 2014; Higuchi et al, 2014). The concept of mRNA distribution along microtubules appears to be widespread, and has also been observed in highly differentiating muscle cells (Denes et al, 2021). Furthermore, active mRNP shuttling is also known to occur in dendrites and axons participating in distributing mRNAs in neurites (Cajigas et al, 2012; Das et al, 2019; De Pace et al, 2024; Shigeoka et al, 2016; Tushev et al, 2018; Zappulo et al, 2017).

The function of endosomal transport of specific mRNA sets became evident by studying Rrm4 cargo mRNAs. Analysing functional categories of cargo mRNAs containing functionally important binding sites revealed distinct mRNA regulons involved in polarity determination, septin assembly, cell wall remodeling, as well as mitochondrial physiology (Fig. 1B; Olgeiser et al, 2019; preprint: Stoffel et al, 2025b), consistent with the key functions of hyphal growth (see above).

The transport of septins is currently best studied combining in vivo UV crosslinking approaches and RNA live imaging, which revealed that all four septin mRNAs: Cdc3, Cdc10, Cdc11, and Cdc12, are Rrm4-dependent cargo mRNAs (König et al, 2009; Olgeiser et al, 2019; Zander et al, 2016). Interestingly, all four septin translation products as well as ribosomes were present on endosomes, providing the first evidence of endosome-coupled translation (Baumann et al, 2014). This mode of translation is needed for the assembly of defined heteromeric septin complexes on the surface of endosomes. The resulting complexes are transported to the growing apex to form higher-order septin filaments with characteristic gradients emanating from the hyphal tip (Fig. 1B; Baumann et al, 2014; Müntjes et al, 2021; Zander et al, 2016). In accordance, loss of Rrm4 causes detachment of septin mRNAs, encoded proteins, and ribosomes. Furthermore, the gradient in higher-order filaments was no longer formed. Instead, septin subunits built abnormal ring complexes in the cytoplasm (Baumann et al, 2014; Zander et al, 2016). Thus, one crucial

function of endosomal mRNA transport is loading of motile endosomes with newly synthesized proteins, promoting assembly of heteromeric complexes and their delivery to defined subcellular destinations (Müntjes et al, 2021). This is particularly intriguing because the Rab5a-positive transport endosomes also function as early endosomes in endocytosis. Thus, endosomal hitchhiking and local membrane-coupled translation offer a new level of complexity for membrane trafficking, potentially linking endosome-coupled de novo synthesis with recycling of membrane-associated proteins during endocytosis.

The observation of co-transport and membrane-coupled translation of septin subunits is in line with the eukaryotic RNA operon model for assembly. Monocistronic mRNAs of eukaryotes are spatially organized by the action of RBPs to form joint RNA regulons. Their localized translation enables efficient complex assembly (Blackinton and Keene, 2014; Keene, 2007). This regulatory process is widespread and has been shown for the assembly of multi-subunit enzymes and translation initiation factor complexes (Schwarz and Beck, 2019; Shiber et al, 2018). Importantly, endosome-coupled translation is common in eukaryotes. In axons of retinal ganglion cells, local translation at Rab7-positive late endosomes promotes protein targeting to adjacent mitochondria (Cioni et al, 2019; see below). Furthermore, proteins such as the human FYVE protein EEA1 are co-translationally targeted to endosomes (preprint: Popovic et al, 2020).

In other fungi, the knowledge of the biological functions of vesicle-mediated mRNA transport is less detailed. However, the Rrm4-dependent machinery from Basidiomycota appears to be widespread, ranging from Chytridiomycota and Mucoromycota (Müller et al, 2019). Consistently, the evolutionarily conserved translational repressor Ssd1p/GUL1 and the poly(A) binding protein Pab1 shuttle on endosomes (GUL1 constitutes the Ssd1p homolog from *Neurospora crassa* and *Sordaria macrospora;* Hall & Wallace, 2022; Herold et al, 2019; Stein et al, 2020). A prominent function of Ssd1p/GUL1 is the regulation of mRNAs involved in cell wall remodeling (Herold et al, 2019; Jansen et al, 2009; Kurischko et al, 2011; Stein et al, 2020), supporting the observation that the Rrm4-dependent endosome trafficking machinery transports RNA regulons encoding cell wall synthetic enzymes (preprint: Stoffel et al, 2025b). Ssd1p/GUL1 constitutes a target of the evolutionarily conserved signaling complex STRIPAK "striatin-interacting phosphatase and kinase" involved in various diseases, cancer, and cardiovascular disorders (Kück et al, 2019; Stein et al, 2020). The STRIPAK-dependent phosphorylation status might regulate the translation status of cargo mRNAs during transport. In humans, the STRIPAK complex has been hypothesized to modulate dynein activity in axons and might thereby influence the transport of vesicles and autophagosomes (Neisch et al, 2017). This provides new hints on how posttranslational regulation might influence endosomal mRNP transport (see outstanding questions Box 2).

## Biological functions of vesicle-mediated mRNA transport in higher eukaryotes

In plants, endosomal mRNA transport has been implicated in the regulation of seed and flower development. Transport of mRNAs encoding storage proteins such as glutelin to specific cortical ER subdomains is essential for seed development in *Oryza sativa*. Rab5

mutants (glub4) and Rab5 GEF (glub6) mis-localize glutelin-encoding mRNAs, suggesting the involvement of endosomal transport in plants (Tian et al, 2020a). During flowering, rotamase cyclophilins (ROCs) from *Arabidopsis thaliana* function as noncanonical RBPs on the surface of Rab5-positive endosomes to target FLOWERING LOCUS (FT) mRNA to plasmodesmata for cell-to-cell passage *in planta*. ROC mutants are affected in FT mRNA transport and exhibit aberrant timing of flowering (Luo et al, 2024). Thus, endosomal mRNA transport is operational in plants, but it is currently unclear how these single-domain cyclophilins are attached to endosomes (Table 1).

In vertebrates, most examples of vesicle-mediated mRNA transport are known from highly polarized neuronal cells. Loss of transport has been implicated in neuronal diseases such as amyotrophic lateral sclerosis (ALS), Charcot-Marie-Tooth type 2B, or early-infantile neurogenerative disorders (Liao et al, 2019; Cioni et al, 2019; De Pace et al, 2024; see below). Molecular functions include regulation of polarity, mitochondrial physiology, or response to external cues (Cioni et al, 2019; Corradi et al, 2020; De Pace et al, 2024; Table 1).

Vesicles hitchhiking promotes the transport of two different types of RNA: mRNAs and regulatory RNAs. The latter was uncovered during the directed growth of axons of retinal ganglion cells from *X. laevis* (Corradi & Baudet, 2020; Corradi et al, 2020). Using the tetraspanin CD63 as a marker for late endosomal/lysosomal vesicles (Pols and Klumperman, 2009) revealed that precursors of micro-RNAs (pre-miRNAs) hitchhike on vesicles along microtubules for long-distance transport towards the growth pole (Corradi et al, 2020). Upon exposure to the axon guidance cue semaphorin 3 A, distinct pre-miRNAs are processed to active versions that silence translation of target mRNAs encoding, for example, tubulin beta 3 class TUBB3. This disclosed an essential regulatory process for steering of growth cones and, more broadly, for formation of neuronal circuits (Corradi and Baudet, 2020; Corradi et al, 2020).

Besides pre-miRNA translocation, endosomes also transport distinct mRNAs. β-actin mRNA has been described as a component of higher-order mRNP granules, transported as biomolecular condensates on lysosomes in rat and zebrafish neurons by the ANXA11 machinery (Liao et al, 2019). β-actin mRNA also co-purifies with the neuronal FERRY complex, indicating alternative endosomal transport machineries (Schuhmacher et al, 2023). Since also mRNAs encoding small GTPases were found to be transported, such as RAC1 involved in determining polarity, endosomal transport might be needed to organize the actin cytoskeleton. This is consistent with mRNA targets found in *U. maydis* (preprint: Stoffel et al, 2025b), proposing the recurring theme of a widespread mechanism operational during polar growth in hyphae and neurons. Other sets of prominent cargoes are mRNAs encoding mitochondrial proteins enriched for components of the electron transport chain (ETC; Cioni et al, 2019; De Pace et al, 2024; Olgeiser et al, 2019; preprint: Stoffel et al, 2025b), suggesting a crucial link between endosomes and mitochondria. Therefore, we will focus on this aspect in the remainder of the review.

## Posttranscriptional control of mitochondrial physiology

About 99% of the ~1100 mitochondrial proteins in mammals are encoded by the nuclear genome and translated in the cytoplasm for mitochondrial protein import using defined protein targeting sequences (Kaushik et al, 2025; Pfanner et al, 2019). The process must be executed efficiently and accurately, since inefficient protein import leads to mitochondrial dysfunction. Mislocalisation of mitochondrial precursor proteins might result in their aberrant cytoplasmic aggregation, a process linked to disease and aging (Kramer et al, 2023). Efficient protein entry is promoted by local translation of cognate mRNAs in the vicinity of mitochondria (Fazal et al, 2019; Williams et al, 2014; Fig. 3). Protein import might even take place co-translationally, since cytosolic 80S ribosomes are physically associated with the outer mitochondrial membrane (Kellems et al, 1974; Kellems and Butow, 1972), often with their exit tunnel oriented towards the organelle (Gold et al, 2017). Evidence for active translation was provided by proximity-based ribosome profiling experiments (Vardi-Oknin and Arava, 2019; Williams et al, 2014). Thereby, the nascent peptide chain containing the N-terminal mitochondrial targeting sequence might interact with cognate receptors such as Tom20 (Eliyahu et al, 2010). Alternatively, outer membrane protein OM14 might serve as a receptor for nascent chain-associated complex (NAC) during translation to anchor mRNAs to mitochondria (Lesnik et al, 2014).

A distinct mechanism to promote translation in the vicinity of mitochondria is the action of RBPs, which specifically recognize RNA elements as targets for mitochondrial anchoring. Puf3p from *S. cerevisiae*, for example, binds to the 3′-UTR of numerous mRNAs encoding mitochondrial proteins and thereby promotes their mitochondrial protein import (Bykov et al, 2020; Eliyahu et al, 2010; Gerber et al, 2004; Saint-Georges et al, 2008; Fig. 3). Its activity is regulated by the phosphorylation by nutrient-responsive kinases (Lee and Tu, 2015). Notably, the list of RBPs with functions in mitochondrial-associated mRNA control is steadily growing, including examples such as AKAP1 (protein A-kinase anchoring protein 1) and LARP4 (protein A-kinase anchoring protein 1; La-related protein 4, respectively; Lewis et al, 2024; Luo et al, 2025; Sharma and Fazal, 2024). A substantial step forward in understanding the logic of mitochondrial-coupled translation was achieved by the recent establishment of optogenetic regulation of local ribosome proximity labeling (LOCL-TL, LOV-domain-controlled ligase for translation localization, Luo et al, 2025). Applying this minimal invasive approach revealed that under physiological conditions, one-fifth of the human nuclear-encoded mitochondrial mRNAs are translated on the outer mitochondrial membrane. mRNA association is mediated by the RBP AKAP1, and encoded proteins are enriched for ETC components (Luo et al, 2025).

The diversity of RBPs and their targets suggests a highly modular system of control. Rather than up- or down-regulating the entire mitochondrial proteome, the cell can use specific RBPs to fine-tune distinct functional modules, such as ETC respiration (Lewis et al, 2024; Luo et al, 2025). This provides additional regulatory flexibility, allowing for tailored adjustments to mitochondrial function in response to specific physiological demands.

## Linking vesicle-mediated mRNA transport with mitochondrial physiology

An outstanding key question is: how do mRNAs reach mitochondria? This is particularly important for mitochondria distant from

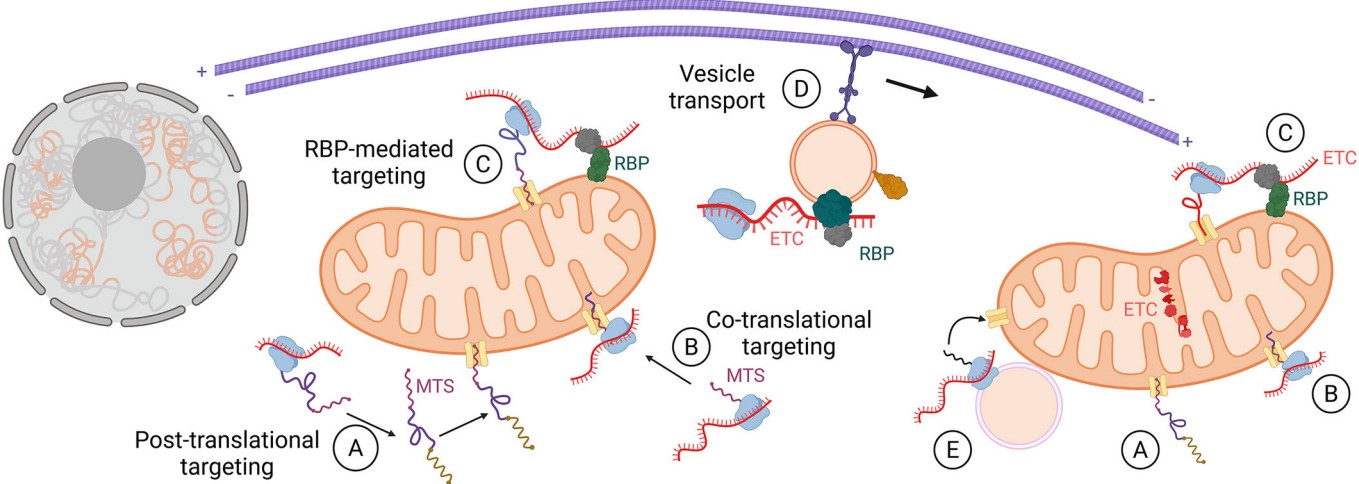

**Figure 3.  Long-distance transport of electron transport chain encoding mRNAs supply distant mitochondria with proteins.**

mRNAs for mitochondrial proteins are transcribed in the nucleus and exported to the cytoplasm. For mitochondria near the nucleus, protein import occurs via several pathways: (**A**) posttranslational import: proteins are synthesized on free ribosomes before being imported; (**B**) co-translational import: translation is coupled to entry at the mitochondrial surface; and (**C**) RBP-mediated tethering: RNA-binding proteins anchor specific mRNAs to the mitochondrial outer membrane to facilitate local translation and import. (**D**) To supply mitochondria at distant cellular locations, such as in axons or fungal hyphae, a specialized mechanism is required. mRNAs encoding key proteins like those of the electron transport chain (ETC) are transported long distances by hitchhiking on vesicles that move along microtubules, ensuring efficient on-site protein delivery to maintain mitochondrial homeostasis throughout the cell. (**E**) Local translation at late endosomes promotes entry of distinct mitochondrial proteins (further details are given in the text; created in https://BioRender.com).

the nucleus in highly polarized cells such as fungal hyphae and neurites (Cioni et al, 2019; De Pace et al, 2024; Olgeiser et al, 2019; Fig. 3). mRNA transport might result in local mitochondrial activity at cellular outposts, offering a potential subcellular hotspot of ATP production. In fungi, this could be crucial to serve the increased energy demand of the highly active growth pole.

In *U. maydis*, we observed that mRNAs encoding mitochondrial proteins are prominent cargos of Rrm4 (Olgeiser et al, 2019; Stoffel et al, 2025a; preprint: Stoffel et al, 2025b). mRNAs encoding subunits of all five mitochondrial electron transport chain complexes were found to be Rrm4 in vivo targets, suggesting a link to mitochondrial oxidative phosphorylation. Furthermore, a detailed analysis of the RRM domain-dependent RNA binding capacity of Rrm4 revealed functionally important binding sites in mitochondrial targets such as the $F_1F_O$ ATP synthase subunits Atp4 or Atp5 (preprint: Stoffel et al, 2025b). Consistently, an earlier proteome study of membrane-associated fractions from hyphae revealed that loss of Rrm4 reduces the accumulation of membrane-bound Atp4 (Koepke et al, 2011). Hence, endosomal mRNA transport might be needed to transport numerous mRNAs encoding mitochondrial proteins, and some of them, like Atp4, exhibit a very important target (VIT) status. Importantly, this is in line with the LOC-TL results reporting that human ETC components use the mitochondria-coupled translation pathway for entry (Luo et al, 2025).

Thus, endosomal mRNA transport might be important for mitochondrial protein import to orchestrate entry of subunits, thereby regulating respiration and subcellular metabolism. Consistent with this view, we discovered that *rrm4Δ* hyphae become more sensitive to the ATP synthase inhibitor oligomycin, suggesting altered mitochondrial metabolism.

In neurons, local mitochondrial activity has been implicated to be crucial for neuronal activity. One prominent link between RNA transport and mitochondria is mRNA hitchhiking on the surface of mitochondria (Box 1). Alternatively, in axons of retinal ganglion cells from *X. laevis*, endosome-coupled translation on the surface of Rab7-positive late endosomes is important for local translation of mitochondrial proteins. This enables efficient targeting to mitochondria in the vicinity of late endosomes, critical for correct mitochondrial homeostasis (see below; Cioni et al, 2019). Furthermore, inhibiting kinesin-dependent long-distance transport by knockout of the lysosome-kinesin adapter BORC (BLOC1-related complex) turned out to be very informative (De Pace et al, 2024). Loss of lysosome-related vesicle transport caused depletion of a specific mRNA set in mammalian axons, causing their degeneration. mRNAs encoding ribosomal and mitochondrial proteins were identified, and loss of transport altered mitochondrial morphology and physiology (De Pace et al, 2024). In essence, vesicle-mediated mRNA transport might be a widespread mechanism to orchestrate mitochondrial homeostasis.

## Concluding remarks

mRNA transport by endosomal hitchhiking was initially viewed as a specialized translocation mechanism restricted to fungal hyphae of the plant pathogen *U. maydis*. However, recent results indicate that vesicle-mediated mRNA trafficking is widespread in eukaryotes and found in fungi, plants, animals, and humans. Different types of membranous vehicles exist, including early endosomes, late endosomes and lysosomes (Baumann et al, 2012; Cioni et al, 2019; De Pace et al, 2024; Liao et al, 2019). Accordingly, different translocation machineries, such as Rrm4-, FERRY-, and ANXA11-dependent, are operational (Table 1). Although individual components are not conserved, the underlying fundamental principles are

shared: Key RNA-binding components interact with endosomal linkers for membrane attachment (Devan et al, 2024; Quentin et al, 2023; Schuhmacher et al, 2023). Intriguingly, common sets of cargo RNA regulons such as polarity factors and mitochondrial ETC proteins are shared (De Pace et al, 2024; Olgeiser et al, 2019; Quentin et al, 2023; Schuhmacher et al, 2023; preprint: Stoffel et al, 2025b). Since a subset of transcripts, such as septin mRNAs, is translated during transport, membrane-coupled translation is no longer restricted to the ER and mitochondria but also takes place at vesicular units such as endosomes and lysosomes (Baumann et al, 2012; Cioni et al, 2019).

*U. maydis* has matured into one of the best studied model systems to understand mRNA translocation in general, including key RBPs, cargo RNA regulons, transcriptome-wide cargo recognition and adapter attachment to membranes. A common theme is that multiple domains, i.e., three RRMs for RNA binding and three MLLEs for adaptor attachment, function with a strict hierarchy: RRM1 and RRM2 are crucial for recognition of functionally important binding sites, and MLLE3 is necessary and sufficient for endosomal attachment. The remaining domains (RRM3, MLLE1, and MLLE2) play accessory roles and foster interaction of the main domains, with RNA or protein partners (Devan et al, 2024; preprint: Stoffel et al, 2025b). Some outstanding questions to be addressed to further advance this interdisciplinary research field are listed in Box 2.

# Outlook

Studying vesicle-mediated mRNA transport and local membrane-coupled translation has increased our understanding of the underlying mechanisms in biological processes such as fungal infection, plant development and neuronal functions. Knowledge of the underlying intensive organelle networking provides new insights into the evolutionary trajectory during mitochondrial protein import (Luo et al, 2025). Furthermore, RRM protein-mediated membrane-coupled translation is also involved in local translation at subcellular thylakoid membranes in cyanobacteria with important implications for chloroplast evolution (Hemm et al, 2025; Hess et al, 2025). Finally, the gained knowledge might inform applied science to tackle current challenges such as fighting fungal disease, improving crop plants, and curing neuronal disease. We are also confident that RNA/membrane research will offer new solutions in approaches such as mRNA delivery during vaccination against viral disease or cancer.

# Peer review information

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

## Acknowledgements

We thank lab members for their critical reading of the manuscript. The work was funded by grants from the Deutsche Forschungsgemeinschaft under Germany's Excellence Strategy EXC-2048/2—Project ID 39068111 to MF; DFG-SFB1535 - project ID 458090666 to MF (project A03) and (project Z01) to SHJS; DFG-CSS—project ID 417919780 to SHJS, as well as DFG-RU5116 - project ID 433194101 to MF (project B03).

## Author contributions

**Melissa Vázquez-Carrada**: Visualization; Writing—review and editing. **Sainath Shanmugasundaram**: Writing—review and editing. **Sander H J Smits**: Funding acquisition; Visualization; Writing—review and editing. **Lasse van Wijlick**: Visualization; Writing—original draft; Writing—review and editing. **Michael Feldbrügge**: Conceptualization; Funding acquisition; Writing—original draft; Project administration; Writing—review and editing.

## Disclosure and competing interests statement

The authors declare no competing interests.

