## [Peer Review File · EMBO Reports]

Vesicle-coupled mRNA transport and translation govern intracellular organelle networking

Melissa Vázquez-Carrada, Sainath Shanmugasundaram, Sander Smits, Lasse Wijlick, and Michael Feldbruegge

Corresponding author(s): Michael Feldbruegge (feldbrue@hhu.de)

Review Timeline:

Submission Date:	24th Sep 25
Editorial Decision:	31st Oct 25
Revision Received:	20th Nov 25
Accepted:	25th Nov 25

Editor: Esther Schnapp

Transaction Report:

Dear Michael,

Thank you for the submission of your Review to EMBO reports. We have now received the full set of referee reports that is pasted below.

As you will see, all referees acknowledge that the review is very interesting and timely, which is great. The referees also all have suggestions for how the review could be improved, and I think all suggestions are good and I would like to ask you to address them. Especially referee 1 provides an excellent report. While point 6 of referee 1 and point 1 of referee 2 are certainly valuable and very good suggestions that would further increase the general interest of this review, I am not sure whether you would like or are willing to address these points, i.e. providing a dedicated figure instead of Box 1 and adding peroxisome-coupled RNA transport to the review. I do agree that a more general title for the review is better. I am happy to discuss any referee comments further with you. Please do not hesitate to contact me if you have any questions.

As for timing, would it be possible for you to submit the revised version of the review by the end of November? Please let me know if this works for you. If the review will be ready and edited in early December we should still be able to publish it online this year, which would be helpful, I think.

The review figures look very good to me and do not need to be redrawn by our graphics designer, if this is OK with you? We can use them as they are, and this will also accelerate the publication of your review.

With best wishes,
Esther

Referee #1:

This is an exceptionally comprehensive and well-written review that synthesizes the emerging field of vesicle-associated RNA transport and local translation. The authors successfully link endosomal trafficking, RNA biology, and mitochondrial regulation within a coherent conceptual framework. The article is timely, given the rapidly growing evidence that local translation on endosomal and lysosomal membranes is a conserved principle in eukaryotes. The text is dense but clear, combining fungal, plant, and neuronal perspectives into a unifying model of organelle networking. The scholarship is impressive, and the inclusion of structural, biochemical, and cell-biological findings makes it a valuable reference for the field.

However, several sections could be streamlined or clarified to enhance readability and to balance mechanistic depth with conceptual overview.

Major comments

1. Conceptual focus and structure

The review sometimes oscillates between detailed structural descriptions (e.g., SLIM/MLLE domain interactions) and broader biological implications. While these mechanistic details are valuable, they occasionally obscure the central message. A schematic summary or table comparing the fungal Rrm4/Upa1 system, the mammalian FERRY and ANXA11 complexes, and plant RBP-L machinery would help readers grasp evolutionary parallels at a glance.

2. Evolutionary conservation versus analogy

The manuscript repeatedly uses the term "conserved," although sequence homology between fungal and metazoan transport components is limited. Clarifying whether these systems are mechanistically homologous or functionally convergent would strengthen the evolutionary argument.

3. Integration with existing models of RNA localization

It would be helpful to discuss how vesicle-linked transport complements or replaces canonical RNP-granule-based pathways, particularly in neurons where multiple modes coexist. Briefly positioning endosomal trafficking relative to known axonal RNA-transport mechanisms (e.g., Staufen, FMRP, kinesin-associated granules) would situate this review within the broader landscape of RNA localization.

4. Mitochondrial coupling and functional evidence

The link between endosomal transport and mitochondrial homeostasis is compelling but somewhat speculative. The section could better distinguish established evidence from hypotheses (e.g., metabolic switching in *Ustilago* mutants). A concise figure illustrating how endosome-coupled translation supplies mitochondrial proteins in different systems would be valuable.

5. Recruitment mechanisms

In the section "Recruitment of mRNAs for vesicle-coupled transport," the authors imply that cargo mRNAs are recognized through conserved sequence motifs by a limited set of transport RBPs. However, recent transcriptome-wide analyses (e.g., Chekulaeva, 2024) have questioned the existence of common localization "zip codes," instead proposing mechanisms based on miRNA evasion and local stabilization. Although the authors cite this work, the conceptual contrast between sequence-driven recognition and context-dependent stabilization remains insufficiently developed. I suggest that the authors explicitly discuss how the mechanistic insights from *U. maydis* (Stoffel et al., 2025a,b) fit within this broader framework, emphasizing that Rrm4 exemplifies one well-defined mechanistic solution among multiple evolutionary strategies for selective mRNA recruitment. This would make the section more balanced and conceptually robust.

6. Improper references

The authors overemphasize certain research groups while underrepresenting others, resulting in an imbalanced citation landscape. In particular, the manuscript attributes the discovery of endosome-mediated mRNA hitchhiking to Holt's group (Cioni et al., 2019), whereas the original study focused primarily on endosomal association rather than active transport. Using Cy3-UTP injection, the authors of that work likely labeled predominantly rRNA within RNA granules, and did not directly quantify β -actin mRNA kinetics. As stated in the original paper: "Collectively, these results suggest that, although RNA granules are mostly transported and distributed in axons in an endosome-independent manner, they frequently associate with both early and late endosomes along the axon (Figure S2E)" and "MB tracking revealed that approximately 25% and 35% of β -actin mRNA granules were, indeed, associated with GFP-Rab5a and GFP-Rab7a (Figures 2H and 2J), of which most were static or oscillatory (Figure 2I)." In addition, the review underplays the substantial contributions from the Schwartz and Harbauer groups, whose work elucidated mechanistic aspects of RNA-mitochondria coupling and trafficking. These findings are currently confined to Box 1 but merit integration into the main text and comparison with other systems. A dedicated schematic figure, rather than a boxed summary, would convey the complexity of these mechanisms and their interconnections more effectively. I recommend that the authors revise this section to ensure balanced and accurate referencing of key contributions in the field.

Minor comments

7. The term "import" is, in some contexts, misleading as it implies a translocation into the mitochondria. For instance, in line 288, and 426, "import in" should be replaced by delivery to or targeting to.

8. Ensure consistent use of species names (*Ustilago maydis*, *Arabidopsis thaliana*, *Homo sapiens*).

9. Figure 2 - The current numbering is confusing and makes it difficult to connect the text to the figure. For example, in Fig. 2B it is unclear which elements correspond to the MLE and RRM domains of Rrm4. The authors should adopt a clearer labeling strategy to efficiently indicate the different domains discussed in the text.

10. Figure 3 - The figure should include endosome-mitochondria contact sites, as demonstrated by Holt's lab, to more accurately reflect current evidence.

11. Box 1 should be replaced by a figure.

12. Line 278, The concept of "recycling" appears without context and should be clarified or omitted.

13. Typos: line 504 (to from), Line 99 comma instead of period; the sentence starting in line 231 lacks a verb

Referee #2:

Recent years have witnessed a dramatic change in our way to understand dynamic cell compartmentalization, with two important new concepts arising: i- subcellular compartmentalization can be achieved via membraneous and membraneless organelles, and ii- organelles are in interaction, forming dynamic networks of interdependent entities.

This review focused of a related timely and interesting process in which intracellular transport of mRNAs is promoted through hitchhiking on membraneous organelles, thus enabling regulated spatial RNA localization as well as inter-organelle coupling for translation of organelle-specific proteins. Although a previous review from the authors already addressed this topic (Müntjes et al., 2021, EMBO Reports; PMID: 34402186), important recent articles published since then justifies publishing an updated review on this topic.

Overall, the review is well-written and documented and the authors have made a great effort in summarizing the literature. Figures are also of high quality. As described in my detailed comments below, I however felt that important remodeling is needed before publication. First, the review currently suffers from important imbalances that should be corrected to produce a review of more general interest. Second, the overall structure of the review should be better delineated for improved readability.

1- Is the topic of the review "vesicle-coupled mRNA transport and translation" (as phrased in the current title) or "endosome-

coupled mRNA transport and translation"?

Some major sections of the review cover vesicles in general (eg "basic concepts of organelle-coupled transport..." or "recruitment of mRNAs for vesicle-coupled transport") while some others are specific to endosomes (eg "mechanistic insights into the endosomal association of mRNPs"; "biological functions of endosomal mRNA transport"). I believe it would be of broader interest to cover vesicle-coupled mRNA transport rather than just endosome-coupled mRNA transport (see also point 2). The authors should then also include recent references related to peroxisome-coupled RNA transport and translation (e.g. references from M. Schuldiner's group).

2- The *U. maydis* fungus model, used by the authors, is a great example where the function and mechanisms of RNA-endosome coupling have been dissected. While it is fair that emphasis is overall put on this model, the review currently describes in great details work performed on *U. maydis* and only briefly mentions work from other biological systems. Important recently published work mentioned by the authors, and performed in neurons (and others) should be described in more details (e.g. Schuhmacher et al., 2023, *Molecular Cell*, ; De Pace et al., 2024, *Nature Neurosciences*, Cioni et al., 2029, *Cell*; Liao et al., 2019 ...)

3- The authors describe with great details the identity and domain composition of the linker proteins that connect vesicles and mRNPs. Equally important and timely questions are: how is this connection dynamically modulated? How does this connection impact the phase behavior of RNP complexes/condensates? The authors mention the PTM hypothesis with the example of STRIPAK. Some alternative answers to these questions are also found in Nixon-Abell et al., 2025, *Nat com*; Liao et al., 2019, *Cell*... I suggest to include a section/subsection dedicated to these timely questions.

4- Co-translational targeting of mRNAs is an important point mentioned several times in the text, but never really explained in details. I would suggest to explain this process in the first section (basic concepts of organelle-coupled transport) and to better distinguish translation-dependent and translation-independent processes in the rest of the text.

5- Sections are long and it would help the reader if they would be sub-divided into topical, clearly delineated sub-sections. This is particularly true for the "biological functions of endosomal mRNA transport" section.

Minor points:

- Figure 1A: To my knowledge, LAMP1+ lysosomes are transported along microtubules, not actin filaments as depicted on the scheme.
- Figure 2 B: The meaning of the orange and purple connecting lines is unclear.
- Line 385-391: although the concept of riboregulation is interesting, I would not discuss it as it is not completely related.

Referee #3:

The work by Vazquez-Carrada, Shanmugasundaram et al. provides a comprehensive and well-documented overview of current studies linking endosomal and mRNA trafficking to mitochondrial physiology, highlighting the evolutionary conservation of this mechanism across species. The most recent literature is well cited. While the structure of the text could be reorganized to make the discussion more coherent and impactful, as detailed in the specific comments below, this reviewer is generally supportive for publication and would like the authors to address a few points.

MAIN POINT:

In figure 1: The authors depict lysosome-mediated RNA transport as occurring along the actin cytoskeleton. While this aspect has not been clearly tested, the current assumption is that lysosomes co-move with RNA granules along microtubules for long-range transport, rather than along actin filaments.

It is also important to mention that no evidence has been reported supporting early endosome-mediated RNA transport via the FERRY complex in neurons.

ADDITIONAL POINTS:

1) In the first paragraph, "Basic concepts of organelle-coupled transport and translation of mRNAs", the authors outline the molecular mechanisms through which mRNAs can accumulate at specific subcellular sites. In this context, they appear to oppose the involvement of mRNA stability, referring to the work of Chekulaeva (2024), with motor-dependent mRNA transport. The concept of "stability-driven localization" refers to a regulatory mechanism in which mRNAs localize to specific subcellular regions as a result of their increased stability. However, this does not exclude the possibility that such transcripts can still be recruited to motor proteins through RNA-binding proteins (RBPs) or by hitchhiking on membrane-bound organelles. To avoid possible misinterpretation, I recommend clarifying that these important concepts, which have been discussed in previous studies, are not excluding each other depending, even for the same transcript.

2) At page 9, the authors refer to the recent study by Stoffel et al., (2025b) to support the idea that mRNAs are "protected from degradation during transport." While this reference is relevant, the statement as currently phrased may be somewhat overstated. It appears that the cited study primarily shows that loss of Rrm4 function results in decreased levels of its target mRNAs, suggesting that Rrm4 binding contributes to transcript stability. However, direct evidence that this protective effect specifically occurs, and is functionally linked, to endosomal transport is maybe less clear? Would it be possible that the observed stabilization could be a more general consequence of Rrm4-mRNA interaction, independent of the endosomal transport. Is this mrna binding to rrm4 only at endosomes? Since the cited work is currently available only as a preprint, it would be advisable to present these conclusions with appropriate caution.

3) Page 8, line 210: It is unclear how these 1000 mRNAs have been characterized as cargo in neurons specifically? This sentence sounds confusing.

4) Many details are provided about what is known in fungi, but the authors could also include more information on what has been described in mammals. For instance, which RNA-binding proteins (RBPs) have been found to associate with lysosomes and Annexin A11?

5) The section "Biological functions of endosomal mRNA transport" provides an extensive overview of studies addressing endosomal mRNA transport across different systems. However, despite the impressive amount of information, the text does not deliver a clear and logically organized description of the biological functions underlying these examples. The narrative frequently shifts between model organisms, molecular details, and evolutionary considerations without an explicit conceptual framework, which makes it difficult for the reader to get the main conclusions. If possible, I would encourage the authors to reorganize this section around a few well-defined functional themes to make it more accessible and scientifically informative.

Referee #4:

This review by Vazquez-Carrada et al. summarizes recent advances in our understanding of the mechanisms and functions of mRNA hitchhiking on organelle membranes in polarized cells such as filamentous fungi and neurons. The authors' review is thorough and well written. While the manuscript addresses some aspects of mRNA hitchhiking that have been discussed in previous reviews, their major claims (detailed below) are convincing and novel based on their assessment of research papers that have been published recently within the field.

The authors emphasize that while the protein complexes responsible for attaching and transporting mRNA on organelle membranes differ across species, the underlying principles of recruitment and selection are evolutionarily conserved. Namely, association with RNA binding proteins for specific mRNAs are multivalent, multifactorial, and finely tuned to facilitate localized translation of proteins that are essential for overall cellular function. The authors' focus on the link between mRNA transport on endosomes and mitochondria physiology strengthens their position that mRNA transport is an important mediator for the exquisite interplay and communication between organelles.

The authors' summary of the recent work from the Feldbrügge lab within the broader context of the mRNA hitchhiking field shows the importance of using filamentous fungi as model organisms for polarized cells to gain insights in mechanistic cell biology that are generalizable to plants and neurons. As such, this review will not only be interesting to those who study filamentous fungi, but any who study polarized cells in general.

One minor suggestion that is perhaps beyond the scope of this review, yet I think would be interesting for the authors to discuss in more detail: the dynamics of the mRNP granules upon arriving to their "destination" for local translation. I appreciate that this is alluded to in the points listed in Box 2. However, beyond the STRIPAK complex that acts on Ssd1/GUL1 and is well conserved, are there other master regulators in polarized cells that have been shown to target either RNA binding proteins, or to target mRNA directly for translation initiation/mRNA decay?

Re: Revision of Manuscript # EMBOR-2025-62805V1 "Vesicle-coupled mRNA transport and translation govern intracellular organelle networking"

Dear Editor and Referees,

We gratefully acknowledge the constructive criticism that allowed us to improve our manuscript substantially and we hope that it now merits publication in EMBO Rep.

We have carefully considered all comments and have revised the manuscript accordingly. Please find below a point-by-point response to the referees' comments. Our response is given in blue font.

Referee #1:

This is an exceptionally comprehensive and well-written review that synthesizes the emerging field of vesicle-associated RNA transport and local translation. The authors successfully link endosomal trafficking, RNA biology, and mitochondrial regulation within a coherent conceptual framework. The article is timely, given the rapidly growing evidence that local translation on endosomal and lysosomal membranes is a conserved principle in eukaryotes. The text is dense but clear, combining fungal, plant, and neuronal perspectives into a unifying model of organelle networking. The scholarship is impressive, and the inclusion of structural, biochemical, and cell-biological findings makes it a valuable reference for the field.

However, several sections could be streamlined or clarified to enhance readability and to balance mechanistic depth with conceptual overview.

Major comments

1. Conceptual focus and structure

The review sometimes oscillates between detailed structural descriptions (e.g., SLiM/MLLE domain interactions) and broader biological implications. While these mechanistic details are valuable, they occasionally obscure the central message. A schematic summary or table comparing the fungal Rrm4/Upa1 system, the mammalian FERRY and ANXA11 complexes, and plant RBP-L machinery would help readers grasp evolutionary parallels at a glance.

We thank the reviewer for this constructive feedback regarding the balance between mechanistic detail and broader implications. We agree that a comparative overview would significantly help readers grasp the parallels in RNA/endosome association systems.

To address this:

We have incorporated a new Table 1 that serves as a schematic summary, directly comparing the core components and functions of the fungal (Rrm4/Upa1), mammalian (FERRY and ANXA11 complexes), and plant (RBP-L machinery) systems.

2. Evolutionary conservation versus analogy

The manuscript repeatedly uses the term "conserved," although sequence homology between fungal and metazoan transport components is limited. Clarifying whether these systems are

mechanistically homologous or functionally convergent would strengthen the evolutionary argument.

At present, it is difficult to assess, whether endosomal mRNA transport is homologous or convergent. We do find RRM proteins in all systems that have been described to function in transport. However, the core component Rrm4 is only found in basidiomycetes. For clarification we avoided the term “conserved” and describe that the process is wide-spread and common in eukaryotes.

3. Integration with existing models of RNA localization

It would be helpful to discuss how vesicle-linked transport complements or replaces canonical RNP-granule-based pathways, particularly in neurons where multiple modes coexist. Briefly positioning endosomal trafficking relative to known axonal RNA-transport mechanisms (e.g., Staufen, FMRP, kinesin-associated granules) would situate this review within the broader landscape of RNA localization.

In the chapter “Basic concepts ...” we now introduce the different mechanisms of active transport in the order of increasing complexity. In case of canonical mRNP transport we use the She2-She3/Myo4, the APC/KIF3 and the CNBP/KIF5 examples. The first two have been also analysed using *in vitro* systems and the RNA transport element for CNBP was recently reported. In order to stress the importance of coordinating multiple modes in neurons, we included an additional question in the Box2 “in need for answers”.

(vii) How are different active transport mechanisms such as classical RBP/motor-based systems or vesicle-coupled systems coordinated in the same cellular environment such as in neurons ?

4. Mitochondrial coupling and functional evidence

The link between endosomal transport and mitochondrial homeostasis is compelling but somewhat speculative. The section could better distinguish established evidence from hypotheses (e.g., metabolic switching in *Ustilago* mutants). A concise figure illustrating how endosome-coupled translation supplies mitochondrial proteins in different systems would be valuable.

Following this constructive criticism, we improved this chapter of the manuscript.

- (i) As suggested by Referee #2 we deleted the speculative part of riboregulation.
- (ii) We divided that chapter in two parts introducing a new subheader “Linking vesicle-mediated mRNA transport with mitochondrial physiology”
- (iii) We shorten the aspect on metabolic switching in *Ustilago* mutants. The text now reads “Consistent with this view, we discovered that *rrm4Δ* hyphae become more sensitive to the ATP synthase inhibitor oligomycin, suggesting altered mitochondrial metabolism.”
- (iv) We tone down the final conclusion. “In essence, vesicle-mediated mRNA transport might be crucial to orchestrate mitochondrial homeostasis.”
- (v) As suggested in comment number 10, we included local translation on late endosomes in Fig. 3. This visualizes the difference between mRNA transport on early endosomes and local translation on late endosomes.

5. Recruitment mechanisms

In the section "Recruitment of mRNAs for vesicle-coupled transport," the authors imply that cargo mRNAs are recognized through conserved sequence motifs by a limited set of transport RBPs. However, recent transcriptome-wide analyses (e.g., Chekulaeva, 2024) have questioned the existence of common localization "zip codes," instead proposing mechanisms based on miRNA evasion and local stabilization. Although the authors cite this work, the conceptual contrast between sequence-driven recognition and context-dependent stabilization remains insufficiently developed. I suggest that the authors explicitly discuss how the mechanistic insights from U. maydis (Stoffel et al., 2025a,b) fit within this broader framework, emphasizing that Rrm4 exemplifies one well-defined mechanistic solution among multiple evolutionary strategies for selective mRNA recruitment. This would make the section more balanced and conceptually robust.

As pointed out in the chapter "basic concepts of organelle-coupled transport and translation of mRNAs" we differentiate between stability-driven localisation and active transport. The work by the Chekulaeva lab has nicely demonstrated that localisation and local translation can be mediated by miRNA evasion and local stabilisation. Here, *cis*-acting RNA elements within the target mRNAs contain most likely instability elements. However, in this chapter we would like to focus on the question of how mRNAs are recruited for active vesicle-coupled transport.

Work from other systems such as She2/She3 interacting with ASH1 mRNA or CNBP interacting with GA-rich sequences have identified such RNA transport elements within the cargo RNAs that are bound by trans-active factors, i.e. proteins with RNA-binding domains, for canonical mRNA transport of few examples. Our work on Rrm4 has revealed that it uses its RRM domains to discriminate between different sets of hundreds of cargo mRNAs. We observe a simple binary interaction, i.e. RRM3 recognizes the short motif UAUG present in numerous mRNAs. This interaction is likely used for bulk mRNA transport. In contrast, the interaction of RRM1 and RRM2 with their cognate binding sites is used to identify functional important binding site found in targets with functions needed for polar growth. In both cases, the binding sites would classify as RNA transport elements. At present, knowledge in the other systems about sequence-specific recognition of cargo mRNAs is not known. It might be that in neurons bulk mRNA transport mediated by RNA granules is coupled to stability-driven localisation.

To clarify that we focus in this chapter on RNA recruitment for active transport we replaced the term RNA localisation element and introduce the term RNA transport element.

At the end of the chapter, we discuss that Rrm4 is one example among other possibilities the text now reads:

"Taken together, uncovering how Rrm4 uses its three RRM domains to recognize different types of RNA transport elements exemplifies one well-defined example of how the translocation of hundreds of mRNAs is coordinated. However, it is conceivable that in other systems, such as neurons, RNA granule-mediated global trafficking of cargo mRNAs lacking specific RNA transport elements is coupled to localisation-driven stability (Chekulaeva, 2024) to determine precise subcellular localisation of mRNAs."

This also addresses point 1 of Referee #3 (see below).

6. Improper references

The authors overemphasize certain research groups while underrepresenting others, resulting in an imbalanced citation landscape. In particular, the manuscript attributes the discovery of endosome-mediated mRNA hitchhiking to Holt's group (Cioni et al., 2019), whereas the original study focused primarily on endosomal association rather than active transport. Using Cy3-UTP injection, the authors of that work likely labeled predominantly rRNA within RNA granules, and did not directly quantify β -actin mRNA kinetics. As stated in the original paper: "Collectively, these results suggest that, although RNA granules are mostly transported and distributed in axons in an endosome-independent manner, they frequently associate with both early and late endosomes along the axon (Figure S2E)" and "MB tracking revealed that approximately 25% and 35% of β -actin mRNA granules were, indeed, associated with GFP-Rab5a and GFP-Rab7a (Figures 2H and 2J), of which most were static or oscillatory (Figure 2I)."

We apologize for this mistake. We now cite the important work from Liao et al. 2019 and de Pace et al. 2024 for their work on vesicle hitchhiking. In addition, we checked the key publications advancing the field substantially. The work by Schumacher et al. 2023, Quentin et al. 2023; De Pace et al. 2024, Cioni et al. 2019 and Liao et al. 2019 is cited in comparable frequency throughout the text.

In addition, the review underplays the substantial contributions from the Schwartz and Harbauer groups, whose work elucidated mechanistic aspects of RNA-mitochondria coupling and trafficking. These findings are currently confined to Box 1 but merit integration into the main text and comparison with other systems. A dedicated schematic figure, rather than a boxed summary, would convey the complexity of these mechanisms and their interconnections more effectively. I recommend that the authors revise this section to ensure balanced and accurate referencing of key contributions in the field.

The topic of hitchhiking on the mitochondrial surface is very interesting and therefore we dedicated Box I to it. However, since we focus in this review on vesicle-mediated mRNA transport we believe that providing this information in Box I is more appropriate.

Minor comments

7. The term "import" is, in some contexts, misleading as it implies a translocation into the mitochondria. For instance, in line 288, and 426, "import in" should be replaced by delivery to or targeting to.

We corrected this.

8. Ensure consistent use of species names (*Ustilago maydis*, *Arabidopsis thaliana*, *Homo sapiens*).

We checked this for *Ustilago maydis* and corrected this for *Arabidopsis thaliana* and *Oryza sativa* (rice). However, for proteins from *Homo sapiens*, *Rattus norvegicus* and *Danio rerio* we would like to stick to the nomenclature "human, rat and zebrafish proteins", because it is more accepted in the field of research. In Table 1 we applied the latin nomenclature.

9. Figure 2 - The current numbering is confusing and makes it difficult to connect the text to the figure. For example, in Fig. 2B it is unclear which elements correspond to the MLLE and RRM domains of Rrm4. The authors should adopt a clearer labeling strategy to efficiently indicate the different domains discussed in the text.

We agree with the reviewer that the original numbering/labeling was confusing. We have revised Figure 2B by enlarging the schematics for Rrm4 and Pab1 to better display their modular organization. Additionally, we have adopted a clearer, domain-based labeling strategy. The Rrm4 domains are now distinctly labeled as RRM1, RRM2, RRM3, and MLLE1, MLLE2, MLLE3, making it unambiguous which elements correspond to which domain (e.g., the RRM vs. MLLE domains). These changes should now efficiently indicate the different domains discussed in the text.

10. Figure 3 - The figure should include endosome-mitochondria contact sites, as demonstrated by Holt's lab, to more accurately reflect current evidence.

We agree with the reviewer's point regarding the importance of inter-organelle contacts. To ensure that Figure 3 accurately reflects current evidence and the work by Holt's lab (and others), we have revised the schematic to explicitly include and highlight the endosome-mitochondria contact sites. The new version of Figure 3 now visually represents this crucial aspect of organelle interaction, improving the accuracy and context of the model presented.

11. Box 1 should be replaced by a figure.

As pointed out above, we would like to keep Box I.

12. Line 278, The concept of "recycling" appears without context and should be clarified or omitted.

We omitted the concept of recycling.

13. Typos: line 504 (to from), Line 99 comma instead of period; the sentence starting in line 231 lacks a verb

We corrected this.

Referee #2:

Recent years have witnessed a dramatic change in our way to understand dynamic cell compartmentalization, with two important new concepts arising: i- subcellular compartmentalization can be achieved via membraneous and membraneless organelles, and ii- organelles are in interaction, forming dynamic networks of interdependent entities. This review focused of a related timely and interesting process in which intracellular transport of mRNAs is promoted through hitchhiking on membraneous organelles, thus enabling regulated spatial RNA localization as well as inter-organelle coupling for translation of organelle-specific proteins. Although a previous review from the authors already addressed this topic (Müntjes et al., 2021, EMBO Reports; PMID: 34402186), important recent articles

published since then justifies publishing an updated review on this topic.

Overall, the review is well-written and documented and the authors have made a great effort in summarizing the literature. Figures are also of high quality. As described in my detailed comments below, I however felt that important remodeling is needed before publication. First, the review currently suffers from important imbalances that should be corrected to produce a review of more general interest. Second, the overall structure of the review should be better delineated for improved readability.

1- Is the topic of the review "vesicle-coupled mRNA transport and translation" (as phrased in the current title) or "endosome-coupled mRNA transport and translation"?

Some major sections of the review cover vesicles in general (eg "basic concepts of organelle-coupled transport..." or "recruitment of mRNAs for vesicle-coupled transport") while some others are specific to endosomes (eg "mechanistic insights into the endosomal association of mRNPs"; "biological functions of endosomal mRNA transport"). I believe it would be of broader interest to cover vesicle-coupled mRNA transport rather than just endosome-coupled mRNA transport (see also point 2). The authors should then also include recent references related to peroxisome-coupled RNA transport and translation (e.g. references from M. Schuldiner's group).

The key aspects of this review is transport and translation of mRNA on vesicles and we discuss endosomes and lysosomal vesicles as prime examples. Therefore, we adjusted the titles of the subsections accordingly. We mention mitochondrial hitchhiking in Box I. In our opinion, peroxisome-coupled translation is beyond the scope of this review.

2- The *U. maydis* fungus model, used by the authors, is a great example where the function and mechanisms of RNA-endosome coupling have been dissected. While it is fair that emphasis is overall put on this model, the review currently describes in great details work performed on *U. maydis* and only briefly mentions work from other biological systems. Important recently published work mentioned by the authors, and performed in neurons (and others) should be described in more details (e.g. Schuhmacher et al., 2023, Molecular Cell, ; De Pace et al., 2024, Nature Neurosciences, Cioni et al., 2029, Cell; Liao et al., 2019 ...)

We fully agree with the reviewer that our review should provide a more balanced and detailed description of RNA-endosome coupling across different biological systems, rather than placing an overwhelming emphasis on the *Ustilago maydis* model.

We have extensively revised the manuscript to incorporate greater detail on key findings from other models, particularly in the context of neuronal transport and signaling. This holds true, for example, for the detailed description of:

The biological function of vesicle-mediated transport in neurons.

The regulation of vesicle attachment by the ANXA11 machinery in mammalian systems.

Furthermore, we have ensured that the important and recently published works mentioned by the reviewer (e.g., Schuhmacher et al., 2023; De Pace et al., 2024; Cioni et al., 2019; Liao et al., 2019) are now described in sufficient detail within the relevant sections of the text.

Finally, as the reviewer noted, the updated Table 1 is now significantly more helpful in facilitating a direct comparison of the function and machinery across the different biological systems, including *U. maydis*, neurons, and others.

3- The authors describe with great details the identity and domain composition of the linker proteins that connect vesicles and mRNPs. Equally important and timely questions are: how is this connection dynamically modulated? How does this connection impact the phase behavior of RNP complexes/condensates? The authors mention the PTM hypothesis with the example of STRIPAK. Some alternative answers to these questions are also found in Nixon-Abell et al., 2025, Nat com; Liao et al., 2019, Cell... I suggest to include a section/subsection dedicated to these timely questions.

We agree that this is a timely question. Therefore, we integrate this question in Box 2. Question III now reads: "How is the link between vesicles and mRNPs dynamically modulated? Study key posttranslational modifications in RBPs, changes in calcium homeostasis and lipid identity."

As recommended, we discussed results from Nixon-Abell 2025 and Liao et al 2019 more thoroughly. We included potential regulation by calcium channels, by modulating lipid identity as well as by the interaction partners ALG2 and CALC that determine condensate formation and physical properties of the underlying lipids. The subsection is found at the end of chapter "mechanistic insights into..."

4- Co-translational targeting of mRNAs is an important point mentioned several times in the text, but never really explained in details. I would suggest to explain this process in the first section (basic concepts of organelle-coupled transport) and to better distinguish translation-dependent and translation-independent processes in the rest of the text.

We explain the basics of co-translational targeting in the first chapter the text now reads

"In recent years, it has become evident that transport of mRNAs and their local translation are key processes to support intracellular networking linking RNA and membrane trafficking (Béthune et al, 2019). Local translation at the rough ER (rough, because of the presence of ribosomes) was reported early on (Blobel & Dobberstein, 1975); however, more recently evidence is accumulating that local translation occurs at numerous intracellular membranes such as those of mitochondria, early endosomes, late endosomes and secretory vesicles (Baumann et al, 2014; Christensen & Reck-Peterson, 2022; Cioni et al, 2019). **Thereby, import into organelles, such as, ER or mitochondria as well as co-translational membrane attachment of translation products on the cytosolic surface of organelles is promoted (see below).** A common mechanism to enable translation in the vicinity of membranes is the function of RNA-binding proteins (RBPs) for mRNA delivery and anchoring, facilitated by membrane-associated adaptor proteins."

We have incorporated the clarifying term "endosome-coupled translation" into Figure 1 to visually emphasize this concept early in the review.

5- Sections are long and it would help the reader if they would be sub-divided into topical, clearly delineated sub-sections. This is particularly true for the "biological functions of

endosomal mRNA transport" section.

We followed this helpful advice and included additional subheadings, also in the chapter on biological functions. See also our response to Referee 3 point 5 for more information.

Minor points:

- Figure 1A: To my knowledge, LAMP1+ lysosomes are transported along microtubules, not actin filaments as depicted on the scheme.

We apologize for the mistake and corrected it.

- Figure 2 B: The meaning of the orange and purple connecting lines is unclear.

We included the information in the figure legend the text now reads:

“interactions depending on PAM2 or PAM2-like SLiMs are indicated by purple or orange lines, respectively”

- Line 385-391: although the concept of riboregulation is interesting, I would not discuss it as it is not completely related.

We removed this paragraph.

Referee #3:

The work by Vazquez-Carrada, Shanmugasundaram et al. provides a comprehensive and well-documented overview of current studies linking endosomal and mRNA trafficking to mitochondrial physiology, highlighting the evolutionary conservation of this mechanism across species. The most recent literature is well cited. While the structure of the text could be reorganized to make the discussion more coherent and impactful, as detailed in the specific comments below, this reviewer is generally supportive for publication and would like the authors to address a few points.

MAIN POINT:

In figure 1: The authors depict lysosome-mediated RNA transport as occurring along the actin cytoskeleton. While this aspect has not been clearly tested, the current assumption is that lysosomes co-move with RNA granules along microtubules for long-range transport, rather than along actin filaments.

We thank the reviewer for pointing out this inaccuracy in Figure 1. We apologize for the mistake in depicting the mechanism of lysosome-mediated RNA transport.

We have corrected Figure 1 to align with the current mechanistic understanding. The figure now depicts the long-range co-movement of lysosomes and RNA granules along microtubules, rather than along the actin cytoskeleton. This revision more accurately reflects

the widely accepted model for intracellular transport of these organelles.

It is also important to mention that no evidence has been reported supporting early endosome-mediated RNA transport via the FERRY complex in neurons.

For clarification we include the following sentence: “Based on the observation that the FERRY complex co-localise with early endosomes in neurons, the authors hypothesize a function during endosomal mRNA transport and mitochondrial protein import (Quentin et al, 2023; Schuhmacher et al, 2023; see below).”

ADDITIONAL POINTS:

1) In the first paragraph, "Basic concepts of organelle-coupled transport and translation of mRNAs", the authors outline the molecular mechanisms through which mRNAs can accumulate at specific subcellular sites. In this context, they appear to oppose the involvement of mRNA stability, referring to the work of Chekulaeva (2024), with motor-dependent mRNA transport. The concept of "stability-driven localization" refers to a regulatory mechanism in which mRNAs localize to specific subcellular regions as a result of their increased stability. However, this does not exclude the possibility that such transcripts can still be recruited to motor proteins through RNA-binding proteins (RBPs) or by hitchhiking on membrane-bound organelles. To avoid possible misinterpretation, I recommend clarifying that these important concepts, which have been discussed in previous studies, are not excluding each other depending, even for the same transcript.

To clarify this point we included the statement:

“Noteworthy, the two strategies for mRNA localisation, i.e. either stability-driven or motor-mediated, do not exclude each other. Certain transcripts might be actively transported to subcellular areas of higher mRNA turnover.”

In addition, we mention this aspect at the end of the chapter “Recruitment of mRNAs...” as a response to the Referee 1 point 5 (see above).

2) At page 9, the authors refer to the recent study by Stoffel et al., (2025b) to support the idea that mRNAs are "protected from degradation during transport." While this reference is relevant, the statement as currently phrased may be somewhat overstated. It appears that the cited study primarily shows that loss of Rrm4 function results in decreased levels of its target mRNAs, suggesting that Rrm4 binding contributes to transcript stability. However, direct evidence that this protective effect specifically occurs, and is functionally linked, to endosomal transport is maybe less clear? Would it be possible that the observed stabilization could be a more general consequence of Rrm4-mRNA interaction, independent of the endosomal transport. Is this mRNA binding to rrm4 only at endosomes? Since the cited work is currently available only as a preprint, it would be advisable to present these conclusions with appropriate caution.

We fully agree and phrased the role of Rrm4 in stability and transport more carefully by offering two alternative possibilities. Furthermore, we include the information that a similar hypothesis of protected RNA transport was also proposed by others. The text now reads:

“Thus, either Rrm4 could have dual roles in cytosolic stability and endosomal transport. Alternatively, Rrm4 target mRNAs might be protected from degradation during transport, linking mRNA stability with translocation (Stoffel *et al.*, 2025b). This hypothesis was also proposed in neurons due to the duration of endosome movement over long distances. Consistently, knock out of FERRY subunits resulted in decreased levels of target mRNAs (Schuhmacher *et al.*, 2023).”

3) Page 8, line 210: It is unclear how these 1000 mRNAs have been characterized as cargo in neurons specifically? This sentence sounds confusing.

We agree with the reviewer's concern that the statement regarding the characterization of 1000 mRNAs as specific neuronal cargo was unclear and potentially confusing, as the precise source and methods for this characterization were not adequately detailed in the text.

We have deleted the sentence to avoid confusion and ensure that all remaining statements regarding mRNA cargo numbers are clearly supported by the cited literature.

4) Many details are provided about what is known in fungi, but the authors could also include more information on what has been described in mammals. For instance, which RNA-binding proteins (RBPs) have been found to associate with lysosomes and Annexin A11?

We included the information that RBP components of stress granules such as FUS and hnRNPA3 are found associated with ANXA11-positive lysosomes (Liao *et al.* 2019). However, their specific function is not clear yet.

5) The section "Biological functions of endosomal mRNA transport" provides an extensive overview of studies addressing endosomal mRNA transport across different systems. However, despite the impressive amount of information, the text does not deliver a clear and logically organized description of the biological functions underlying these examples. The narrative frequently shifts between model organisms, molecular details, and evolutionary considerations without an explicit conceptual framework, which makes it difficult for the reader to get the main conclusions. If possible, I would encourage the authors to reorganize this section around a few well-defined functional themes to make it more accessible and scientifically informative.

We are grateful for this suggestion and improved this part substantially. As suggested by referee 2 point 5, we included more subheadings. We divided the chapter in two parts describing the biological functions known from (i) fungi or (ii) higher eukaryotes. The corresponding paragraphs start with the global functions of the transport process before we zoom in on specific examples. Furthermore, we refer to the new Table 1 so that the reader does not get lost. We also provide more detailed information for functions in neurons to balance the information with *U. maydis*.

Referee #4:

This review by Vazquez-Carrada et al. summarizes recent advances in our understanding of the mechanisms and functions of mRNA hitchhiking on organelle membranes in polarized cells such as filamentous fungi and neurons. The authors' review is thorough and well written. While the manuscript addresses some aspects of mRNA hitchhiking that have been discussed in previous reviews, their major claims (detailed below) are convincing and novel based on their assessment of research papers that have been published recently within the field.

The authors emphasize that while the protein complexes responsible for attaching and transporting mRNA on organelle membranes differ across species, the underlying principles of recruitment and selection are evolutionarily conserved. Namely, association with RNA binding proteins for specific mRNAs are multivalent, multifactorial, and finely tuned to facilitate localized translation of proteins that are essential for overall cellular function. The authors' focus on the link between mRNA transport on endosomes and mitochondria physiology strengthens their position that mRNA transport is an important mediator for the exquisite interplay and communication between organelles.

The authors' summary of the recent work from the Feldbrügge lab within the broader context of the mRNA hitchhiking field shows the importance of using filamentous fungi as model organisms for polarized cells to gain insights in mechanistic cell biology that are generalizable to plants and neurons. As such, this review will not only be interesting to those who study filamentous fungi, but any who study polarized cells in general.

One minor suggestion that is perhaps beyond the scope of this review, yet I think would be interesting for the authors to discuss in more detail: the dynamics of the mRNP granules upon arriving to their "destination" for local translation. I appreciate that this is alluded to in the points listed in Box 2. However, beyond the STRIPAK complex that acts on Ssd1/GUL1 and is well conserved, are there other master regulators in polarized cells that have been shown to target either RNA binding proteins, or to target mRNA directly for translation initiation/mRNA decay?

We thank the reviewer for this insightful and timely suggestion. We agree that the dynamics and regulation of mRNP granules upon reaching their local translation destination is a fascinating and crucial area of study, which bridges the scope of our review with cutting-edge research.

While a comprehensive analysis of all regulatory pathways might be beyond the scope of a single review, we have integrated a more detailed discussion on known master regulators that function at the destination site in polarized cells.

Expanded Discussion: We have expanded the text to more thoroughly discuss the concept of destination-specific regulation.

ANXA11 example: As noted, we have already elaborated on the regulation of ANXA11 transport and its function as a membrane-associated adapter (addressing Referee #2, Comment 3). This serves as a key example of a regulator (like ANXA11) that facilitates destination-specific mRNA delivery in systems like neurons.

Focus on Master Regulators: We have specifically added content to discuss other master regulatory complexes (beyond STRIPAK) that are known to target RNA-binding proteins (RBPs) or mRNA decay/translation initiation machinery at the destination membrane, thereby directly addressing the reviewer's query about localized control.

We believe these additions significantly enhance the discussion of granule dynamics and regulatory mechanisms without compromising the review's primary focus.

Prof. Michael Feldbruegge
Heinrich-Heine University Duesseldorf
Institute of Microbiology
Universitaetsstr. 1
Duesseldorf 40225
Germany

Dear Michael,

I am pleased to inform you that your Review has been accepted for publication in EMBO reports. Your manuscript will be processed for publication by EMBO Press. It will be copy edited and you will receive page proofs prior to publication.

When SpringerNature is contacting you to sign the license agreement form please enter this token so that no publication charges will apply:

Token not available
